# CD11c$^{Hi}$ monocyte-derived macrophages are a major cellular compartment infected by *Mycobacterium tuberculosis*

**Jinhee Lee, Shayla Boyce, Jennifer Powers, Christina Baer, Christopher M. Sassetti, Samuel M. Behar** *

Department of Microbiology and Physiological Systems, University of Massachusetts Medical School, Worcester, Massachusetts, United States of America

* samuel.behar@umassmed.edu

## Abstract

During tuberculosis, lung myeloid cells have two opposing roles: they are an intracellular niche occupied by *Mycobacterium tuberculosis*, and they restrict bacterial replication. Lung myeloid cells from mice infected with yellow-fluorescent protein expressing *M. tuberculosis* were analyzed by flow cytometry and transcriptional profiling to identify the cell types infected and their response to infection. CD14, CD38, and Abca1 were expressed more highly by infected alveolar macrophages and CD11c$^{Hi}$ monocyte-derived cells compared to uninfected cells. CD14, CD38, and Abca1 "triple positive" (TP) cells had not only the highest infection rates and bacterial loads, but also a strong interferon-γ signature and nitric oxide synthetase-2 production indicating recognition by T cells. Despite evidence of T cell recognition and appropriate activation, these TP macrophages are a cellular compartment occupied by *M. tuberculosis* long-term. Defining the niche where *M. tuberculosis* resists elimination promises to provide insight into why inducing sterilizing immunity is a formidable challenge.

## Author summary

The initial cell type infected by *Mycobacterium tuberculosis* is generally acknowledged to be the alveolar macrophage; subsequently, the bacilli spread to other types of myeloid cells, and other macrophages, dendritic cells, and neutrophils become infected. These infected cells are recognized by *M. tuberculosis*-specific T cells that restrict bacterial growth. However, *M. tuberculosis* resists elimination and persists as a chronic infection. How *M. tuberculosis* persists despite a robust immune response is a critical question. To identify the cell types where *M. tuberculosis* persists, we combined a comprehensive flow cytometry panel with yellow-fluorescent protein-expressing *M. tuberculosis* to identify the cell types infected and their response to infection. The strong interferon-γ signature of infected cells was consistent with ongoing T cell recognition in the lung. Importantly, we defined a population of myeloid cells, primarily CD11c$^{Hi}$ monocyte-derived cells, which expressed high levels of CD14, CD38, and Abca1, and had the highest infection rates and bacterial loads. Despite evidence of T cell recognition and appropriate activation, these CD11c$^{Hi}$ macrophages remain

**Data Availability Statement:** All gene expression files are available from the GEO database (accession number: GSE146127). All other relevant

data are within the manuscript and its supporting information files.

**Funding:** This study was funded by grants to SMB from the National Institutes of Health (R01 AI106725, R01 AI123286). The funders had no role in study design, data collection and analysis, decision to publish, or preparation of the manuscript.

**Competing interests:** The authors have declared that no competing interests exist.

infected by *M. tuberculosis* long-term. Defining this niche should help answer why *M. tuberculosis* resists elimination from activated macrophages even in the face of T cell immunity.

## Introduction

Our respiratory system is in direct contact with the environment and the lung's large surface area is patrolled by the immune system to eliminate inhaled particles and microbes. Leukocytes, which reside in lung parenchyma and circulate through its extensive vasculature, play crucial roles in combating viruses, bacteria, fungi and parasites. To understand the pathogenesis of these and other diseases, great effort has gone into characterizing the lung's resident myeloid cells [1, 2]. During inflammation, resident cells change phenotype, function, and location, while other cells are recruited and marginate from the blood to enter lung tissue [3, 4]. Myeloid cells have enormous plasticity and their activation states and functions are shaped by exogenous signals to which they are exposed (e.g., bacterial products, cytokines, hypoxia, cigarette smoke) [5–8]. Thus, the same cell type can be pro-inflammatory and resist infection, or anti-inflammatory and help resolve inflammation and remodel injured tissue.

The bacterium *Mycobacterium tuberculosis* (Mtb) causes the human disease tuberculosis (TB). Mtb infects alveolar macrophages (AM) and replicates intracellularly [9]. However, Mtb infects other myeloid lineage cells, including other types of macrophages, dendritic cells (DC), and neutrophils [10–12]. AM appear to poorly control intracellular Mtb growth, while monocyte-derived cells (MDC), such as interstitial macrophages (IM), more efficiently inhibit bacterial growth [13, 14]. Nevertheless, Mtb resists elimination and persists as a chronic infection. The inability of the myeloid cells to contain the infection stems in part from the propensity of infected cells to die a necrotic death leading to bacterial dispersal, and infection of other cells [15, 16]. Ultimately, bacterial growth plateaus only when T cells are recruited to the lung, where they recognize and activate infected cells [17–19]. Indeed, impairment of cell mediated immunity or loss of T cell function increases the risk of developing active TB [20]. However, even in the presence of T cells, viable bacteria persist in animal models of primary TB. Why T cells are unable to mediate sterilizing immunity is unclear although T cell dysfunction and immune evasion are likely contributors.

The difficulty in distinguishing different types of myeloid cells is one barrier to unraveling mechanisms of immune evasion during TB. The inflammatory milieu during infection promotes recruitment, activation and differentiation of myeloid cells in the lung [21]. We developed a gating strategy that unambiguously classifies lung myeloid cells and made use of a strain of H37Rv that constitutively expresses a super-folding yellow fluorescent protein (YFP), to track and sort infected cells. We determined the gene expression profiles of uninfected and infected cells belonging to the three major non-PMN myeloid cell populations. Twenty-five genes were upregulated by all three cell populations and we validated that CD14, CD38, and Abca1 were co-expressed by the majority to infected cells in the lung. These markers were induced by a combination of Mtb and γ-interferon, a product of the T cell response. Here we discovered that these markers identify a subset of CD11c$^{hi}$ monocyte-derived cells (CD11c$^{hi}$ MDC) that is highly infected and defines a cell population where Mtb persists in the lung.

## Results

### Monocyte-derived macrophages increase in the lung parenchyma during Mtb infection

During the past two decades, there have been major advances in our understanding of the origin and differentiation of myeloid cells [2, 5, 9, 13, 22–26]. However, in the lungs of infected

mice, standard classifications can be difficult to implement because of the diverse activation states of different cell types. To overcome this uncertainty, we developed the following flow cytometric strategy to more precisely categorize lung myeloid cells. We first excluded debris, doublets, and dead cells (S1 Fig). CD45$^+$ leukocytes were distinguished from non-hematopoietic cells, a "dump" channel was used to exclude lymphoid cells (T, B, and NK cells), and neutrophils (PMNs) were identified by Ly6g$^{Hi}$ expression (S1 and S2 Figs). AM were identified by their SiglecF$^{Hi}$CD11c$^{Hi}$ phenotype and eosinophils by their intermediate SiglecF expression and lack of CD11c. In uninfected lung, two subsets of conventional CD11c$^{Hi}$ DC were identified based on their differential expression of CD103 and CD11b. Additionally, CD11b$^{Hi}$ cells expressing low or intermediate levels of CD11c were identified as Ly6c$^{Hi}$ or Ly6c$^{Lo}$ monocyte/ macrophages. Pulmonary DC can be identified by their co-expression of CD11c and MHCII in uninfected mice, as only low MHCII levels are expressed by AM (Fig 1A). Similarly, low CD11b levels are expressed by AM (Fig 1B). However, MHCII and CD11b are both significantly upregulated by AM during Mtb infection (Fig 1A and 1B). Importantly, CD11c and SiglecF unambiguously identify AM, even during Mtb infection, and avoids their misclassification as pulmonary DC or recruited myeloid cells (S3 Fig).

We distinguished cells in the lung parenchyma (i.e., tissue resident) from those in the vasculature by intravenously administering anti-CD45 mAb [27] (Fig 1C). In uninfected mice, AM, CD103DC, and most CD11bDC, were tissue resident, while the majority of monocytes, eosinophils and PMNs were circulating (Fig 1D). The intravascular location of the Ly6c$^{Lo}$ population makes it likely that these cells are nonclassical monocytes [1].

After Mtb infection, the absolute number and relative proportion of these populations change and within 3–4 weeks post infection (wpi), there was a significant increase in the total number of myeloid cells in the Mtb-infected lung [10, 21, 28] (Fig 1E). Although the relative abundance of AM declined, they were detected for months following infection (Fig 1F and 1G). The large increase in lung CD11b$^+$CD11c$^+$SiglecF$^-$ myeloid cells after Mtb infection, and their presence in the circulation as well as in the parenchyma led us to conclude that this cell population is heterogenous and includes both CD11bDC and recruited myeloid cells. As we have previously shown that CD11b$^+$CD11c$^+$MHCII$^{Hi}$ myeloid cells in the lungs of Mtb infected mice can be derived from recruited monocytes [21], we favor the idea that most of the CD11b$^+$CD11c$^+$SiglecF$^-$ lung cells are monocyte-derived cells (MDC), but may include resident CD11bDC. Indeed, Norris and Ernst find that there is a large recruitment of monocytes to the lung parenchyma where they proliferate and differentiate into lung resident myeloid cells, including CD11bDC [27]. Based on data presented below, we believe that both the majority of the CD11b$^+$CD11c$^+$MHCII$^{Hi}$ cells are MDC and are more similar to macrophages than DC [13, 21, 27]. Hereafter, we refer to these cells as CD11c$^{Hi}$ MDC. The Ly6c$^{Lo}$ cells that entered the parenchyma are recruited macrophages (RM), another type of monocyte-derived macrophage [1, 2, 24]. Importantly, RM, monocytes, PMN and CD11c$^{Hi}$ MDC were the most abundant myeloid cell types in the lung following Mtb infection (Fig 1F and 1G).

Huang et al identified a population of lung myeloid cells after high-dose intranasal Mtb infection that resembled the population we are calling CD11c$^{Hi}$ MDC [13]. These cells were CD11b$^{Hi}$CD11c$^{int}$MHCII$^{Hi}$ and based on their co-expression of CD64 and MerTK, and lack of SiglecF, these cells were called interstitial macrophages (IM) [2]. To determine whether CD11c$^{Hi}$ MDC were similar to IM, we analyzed lung myeloid cells three weeks after low-dose aerosol Mtb infection using a modified flow panel. The majority CD64$^+$Mertk$^+$ macrophages were AM (~85%), and only 2% of the total myeloid cells in the lungs were IM based on the Huang's definition (Fig 1H). The reciprocal analysis showed that while all AM were CD64$^+$MerTK$^+$, only ~10% of CD11c$^{Hi}$ MDC were MerTK$^+$ (Fig 1I). The monocyte markers Ly6c and CX3CR1, which are often retained by MDC, were expressed by CD11c$^{Hi}$ MDC,

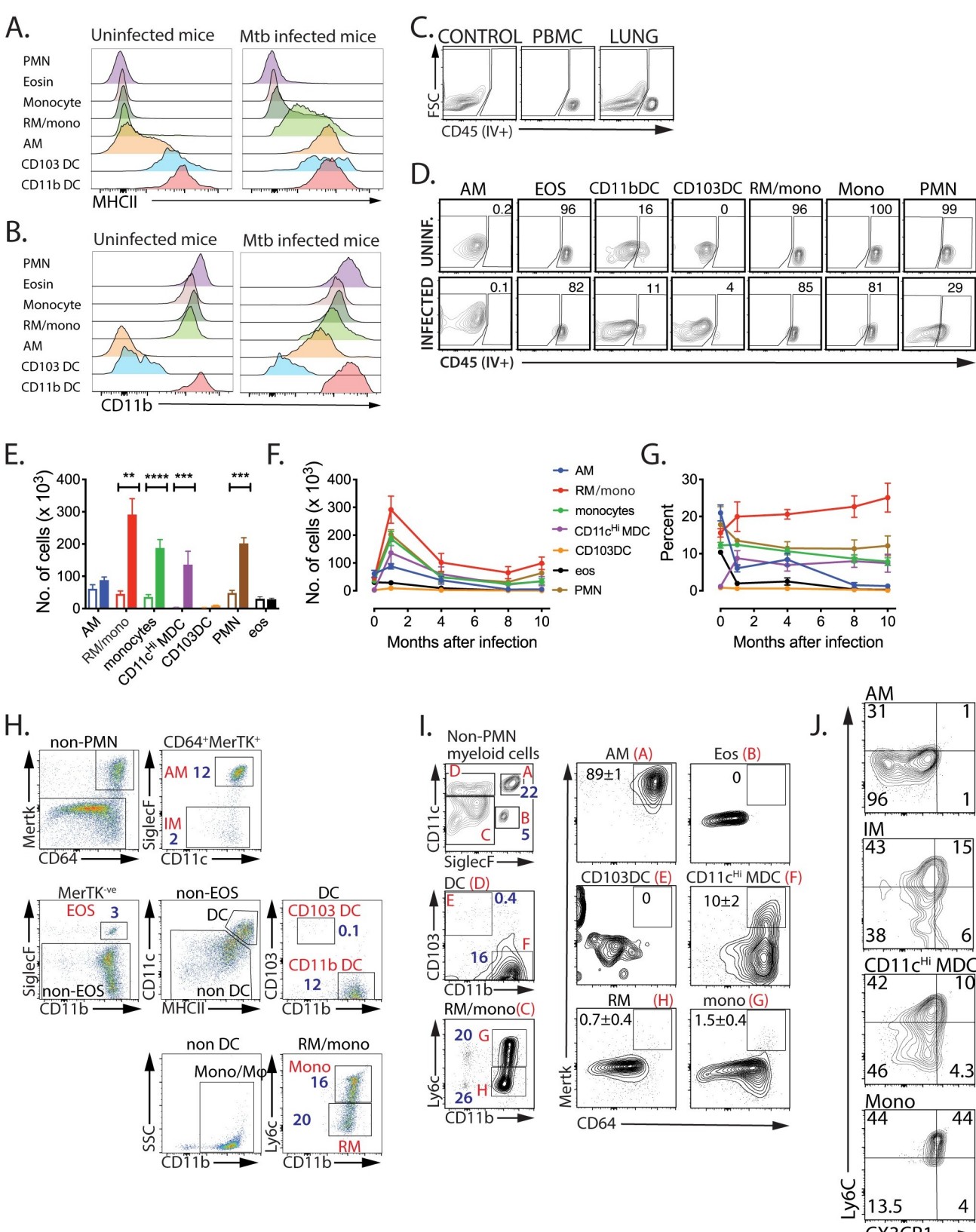

**Fig 1. Expression of MHCII and CD11b are increased by AM during Mtb infection.** A, B. MHCII is significantly upregulated on AM after Mtb infection. C57BL/6 mice were infected by aerosol with ~100 Rv.YFP. After 4 weeks, lung myeloid cells collected from infected mice were compared with those from uninfected mice for the expression of MHCII (A) and CD11b (B) on each myeloid cell subset. C, D. Each myeloid subset populates a distinct anatomical site. Six weeks after infection, mice were injected i.v. with fluorophore-labelled anti-CD45 mAb 3 min before euthanasia. (C, D) "Control" denotes lung cells from an uninjected mouse. PBMC was used to measure the efficiency of vascular staining. Nearly 50% of lung cells were in the vasculature. Each myeloid cell subset was analyzed for its staining with anti-CD45 mAb. E. Myeloid populations selectively expand after infection. Total lung cells were obtained from uninfected mice or mice infected with Rv.YFP for 3 weeks. 3 wpi, the total number of cells was increased in CD11c$^{Hi}$ MDC, RM/mono, monocytes, and PMN after infection, but the number of AM, CD103 DC, and EOS remained unchanged. Each value represents the average of 5 mice. F, G. The number and relative abundance of RM/mono, monocytes, PMNs, and CD11c$^{Hi}$ MDC, peaked 1 month after infection and then declined. Each value represents the average of 5 mice. H: IMs constitute a small fraction of the lung myeloid cells, after 3 weeks post aerosol Mtb infection. The gating scheme used by Huang et al. was used to identify IM (defined as CD64$^+$MertK$^+$SiglecF- non-PMN myeloid cells). The numbers refer to the percentage of each cell population among non-PMN myeloid cells. I. CD11c$^{Hi}$ MDC overlap with IM. Using our gating scheme, we define the percentage of each cell population among non-PMN myeloid cells (left, blue numbers). The expression of CD64 and MertK is determined for each major cell population, and the percentage of CD64$^+$MertK$^+$ cells is indicated by the black numbers. J. AM, IM, CD11c$^{Hi}$ MDC, and Mono were analyzed for the expression of CX3CR1 and Ly6C, markers for monocyte-derived cells. These results are representative of 5 (A-D), 2 (E), 2 (F, G), 2 (H-J) experiments. The SD for **Fig** 1D, 1H, 1I and 1J are not shown for clarity, but were generally <10% of the mean. PMN, neutrophils; EOS or eosin, eosinophils; Mono, monocytes; RM, recruited macrophages, AM, alveolar macrophages; CD103DC, cDC1; CD11bDC, cDC2; CD11c$^{Hi}$ MDC, CD11c$^{Hi}$ monocyte-derived cells. **, p<0.01, ***, p<0.001, p<0.0001, by a two-way ANOVA.

CD64$^+$MerTK$^+$ cells ("IM"), and monocytes, but not AM ([Fig 1J]). The heterogeneity in MerTK expression among CD11b$^+$CD11c$^{Hi}$MHCII$^{Hi}$ cells may reflect distinct cell populations (i.e., DC vs macrophage), or asynchronous recruitment, differentiation or activation. Thus, monocyte-derived macrophages increase in the lung following Mtb infection.

## CD11c$^{Hi}$ MDC are highly infected

Prior studies identified Mtb-infected cells using CD11b and CD11c, focused on one or two cell types, or looked at early time points [10, 13, 21, 28, 29]. We used H37Rv expressing a bright sfYFP (Rv.YFP) [30]. Analysis by image flow cytometry showed that YFP$^+$ cells could be clearly differentiated from the uninfected population. To determine if the MFI of YFP correlated with bacterial burden, we sorted YFP$^{lo}$, YFP$^{int}$, or YFP$^{Hi}$ AM. Imaging of YFP+ cells showed the YFP signal was derived from intracellular fluorescent bacilli. The YFP MFI was proportional to the number of bacilli per cell, showing that MFI could be used as a surrogate for bacterial number ([Fig 2A and 2B]).

Following infection with Rv.YFP, we determined the fraction of lung cells that were infected. At 3 wpi, Rv.YFP$^+$ cells were detected among all seven populations ([Fig 2C]). AM, PMNs, RM, and CD103DC, were infected at a similarly low rate ([Fig 2C]). Remarkably, CD11c$^{Hi}$ MDC were infected at ~20-fold higher rate than AM and other cell types ([Fig 2D]). Thus, even though there are fewer CD11c$^{Hi}$ MDC than PMNs in lung tissue, they become a major intracellular reservoir for Mtb ([Fig 2E]).

## Infected AM have a transcriptional signature signifying interaction with T cells

To assess how different cells respond to intracellular Mtb infection, we transcriptionally profiled infected and uninfected cells from the lungs of Mtb-infected mice 3 wpi. We first analyzed AM as they are the cell type initially infected [9]. Following Rv.YFP infection, YFP$^{pos}$ and YFP$^{neg}$ AM were flow sorted from the lungs of infected mice, as well as AM from uninfected mice (AM$^{Uninf}$). Transcriptional changes were detected in 2,150 genes expressed by YFP$^{pos}$ AM, and 2,392 genes expressed by YFP$^{neg}$ AM, compared to AM$^{Uninf}$, based on an FDR <0.05 and a fold-change ±2-fold ([Fig 3A]). Of the 3,051 genes which changed after Mtb infection, there was an overlap of 1,492 genes, with similar numbers of genes being upregulated or downregulated by YFP$^{pos}$ or YFP$^{neg}$ AM compared to AM$^{Uninf}$ ([Fig 3B and 3C]). These data show that

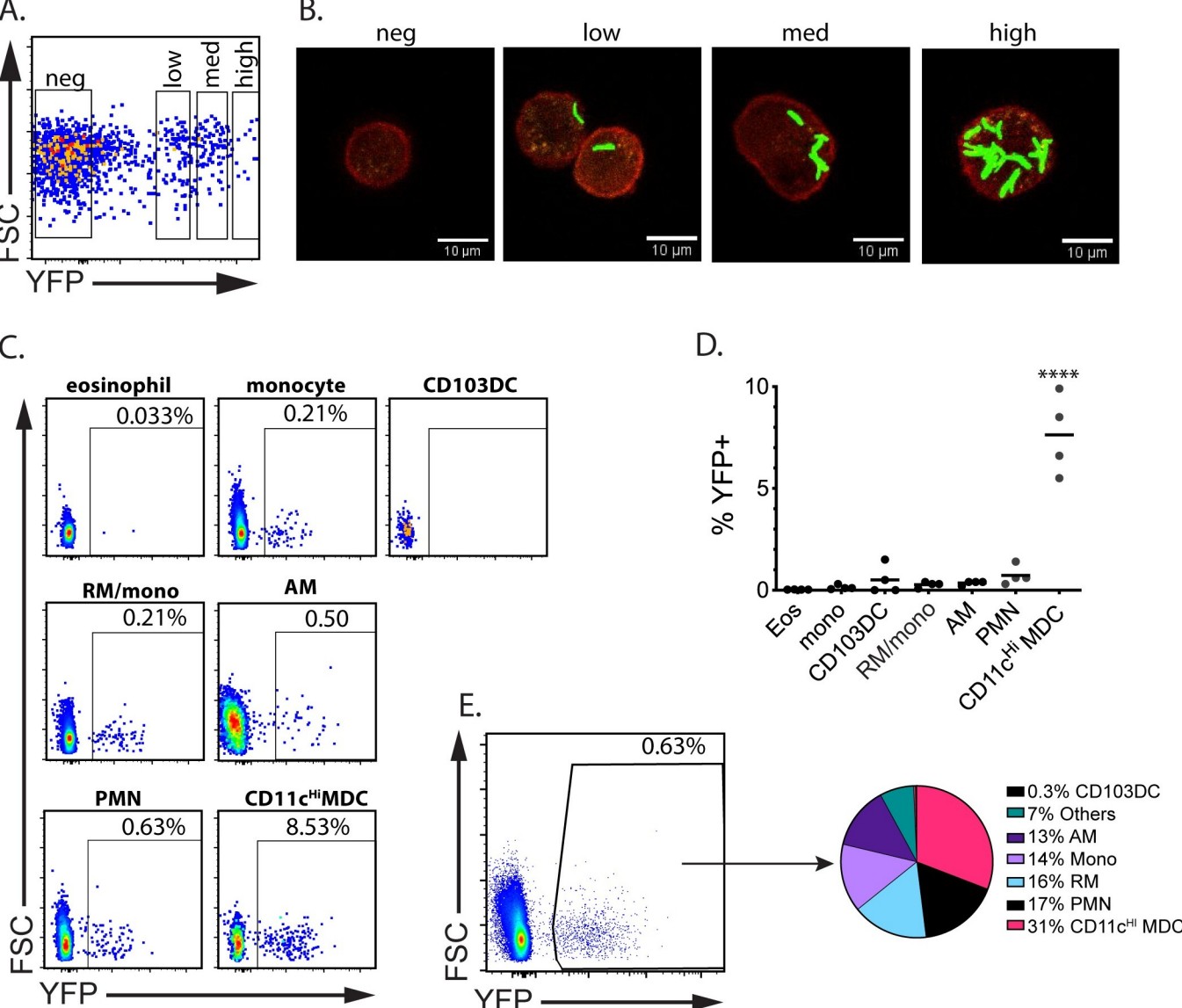

**Fig 2. Infected myeloid cells can be tracked using Rv.YFP.** A, B. To correlate bacterial load and MFI, C57BL/6 mice were infected i.t. with a high dose ($1 \times 10^5$ per animal) of Rv.YFP. Alveolar macrophages were sorted into YFP$^{-ve}$, YFP$^{low}$, YFP$^{int}$, and YFP$^{Hi}$ using a cell sorter (A), and each bin of sorted cells was analyzed by fluorescence microscopy (B). The images were captured by confocal microscopy using a 60x objective. C, D, E. At 3 wpi, total lung cells from infected mice were collected, and each myeloid subset was analyzed to determine the frequency of YFP-positive cells (C). Results from four independent mice summarized (D). From the same experiment, the YFP$^+$ cells were gated and then the frequency of the different myeloid cells present was determined (E). Each pie slice represents the average of 4 mice. These data are representative of 3 experiments. ****, $p < 0.0001$ by 1-way ANOVA compared to eosinophils.

pulmonary Mtb infection affects gene transcription in both uninfected and infected AM, demonstrating how profoundly the inflammatory milieu affects bystander cells.

To identify transcriptional changes that are specific to Mtb-infected AM, we next compared YFP$^{pos}$ AM vs. YFP$^{neg}$ AM. A total of 518 genes were differentially expressed by greater than 2-fold (FDR<0.05), and 370 of these were known genes (Fig 3D). GSEA analysis of the differentially regulated genes (DRG) in YFP$^{pos}$ versus YFP$^{neg}$ AM revealed that several functional pathways were regulated in Mtb-infected AM (Fig 3E). Based on the normalized gene enrichment scores (NES), Mtb-infected AM were activated by IFNγ, TNF and TLR ligands, while cell

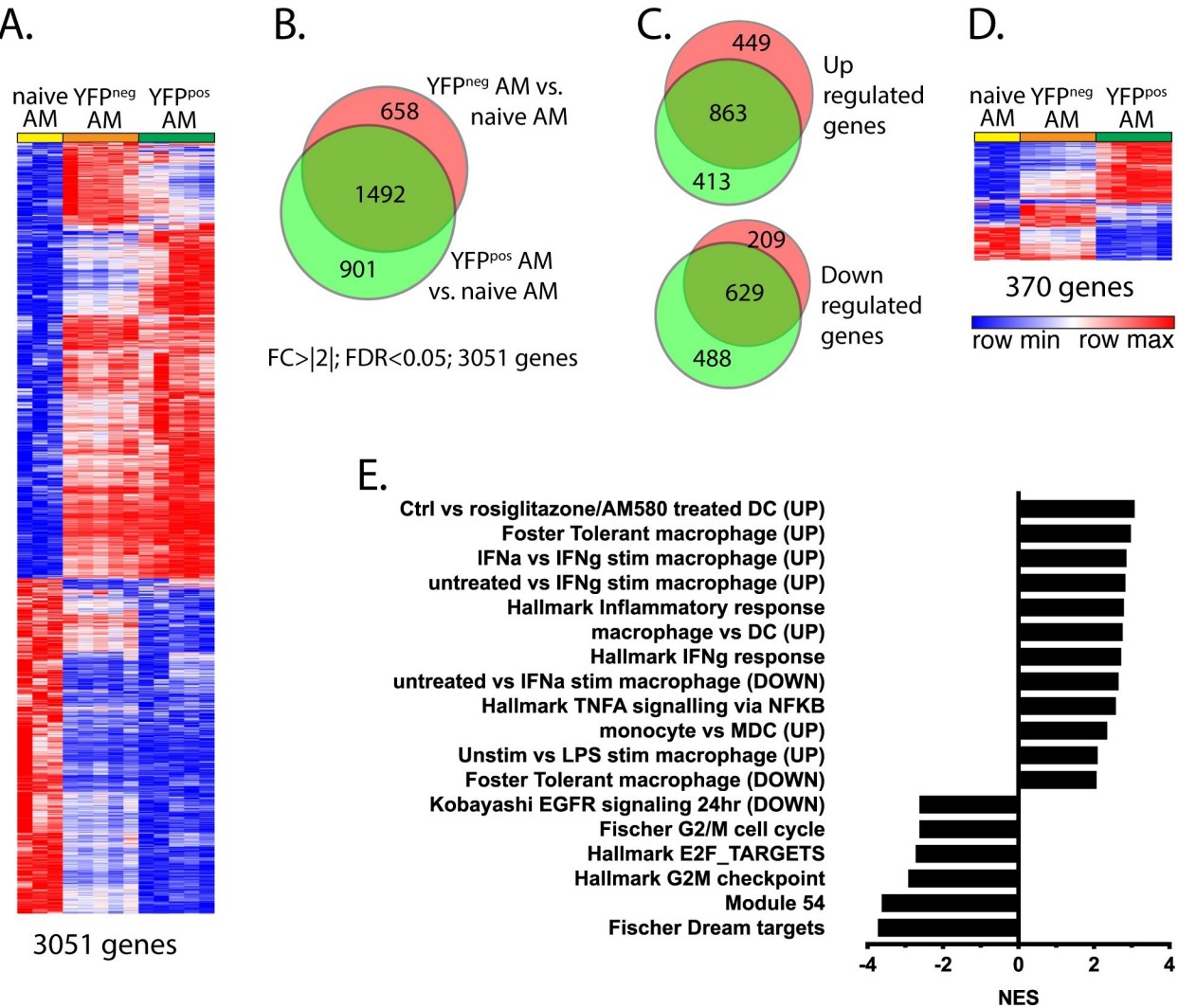

**Fig 3. Infected AM display the unique transcriptional signature.** A. Differentially regulated genes (DRGs) between infected ("YFPpos") or uninfected ("YFPneg") AM from Mtb-infected lungs vs. uninfected macrophages from naïve mice ("N", e.g., YFPpos/N or YFPneg/N) were determined using criteria of [FC]>2 and an FDR<0.05. The 3051 DRGs were clustered and a heat map generated. Each column represents each animal. B. Distribution of genes specifically up or down regulated in AM of Mtb-infected mice vs. AMs of naïve mice were depicted. C. Upregulated genes and downregulated genes were separately analyzed between AM of Mtb-infected mice and AM of uninfected mice. D. 370 genes that were differentially expressed between YFPpos or YFPneg AM from infected mice were clustered, and a heat map generated. E. The results of the GSEA analysis of the 370 DRGs.

division and proliferation were inhibited. Based on the significant enrichment of IFNγ regulated transcripts, we infer that that T cells are specifically interacting with Mtb-infected AM.

## The gene signature of CD11c^Hi MDC is similar to macrophages and monocytes

We next compared the transcriptional profile of YFP^pos AM, YFP^pos CD11c^Hi MDC, and YFP-^pos RM to the transcriptional profile of purified cell types from the lungs of uninfected mice (immgen.org) and to the gene expression signatures previously established for DC, macrophages, and monocytes (Figs 4 and S4) [31, 32]. Mtb-infected AM had a gene expression profile similar to AM (S4A Fig). The YFP^pos CD11c^Hi MDC, which we had originally predicted to

be DC, had a gene expression pattern more consistent with macrophages and distinct from that of AM and DC (Figs 4A and S4A). YFP[pos] RM had a gene signature that overlapped with monocytes and macrophages. These gene signatures were quantitated by determining the relative gene expression for each gene in the signature and comparing the seven populations we analyzed to 43 myeloid cell populations from ImmGen.org (Figs 4B and S4B). Although a general trend was evident, there were some discrepancies even among the ImmGen subsets. For example, CD11b[+] lung macrophages (non-AM) appeared to be more DC-like (Fig 4B). Therefore, we derived a 21 gene signature (S4B and S4C Fig) that distinguishes DC from macrophages and used this signature to perform a Pearson correlation analysis was performed for all cell populations. There was a high degree of similarity between the population of YFP[pos] AM, YFP[pos] CD11c[Hi] MDC, and YFP[pos] RM with normal populations of macrophages. These data are consistent with their derivation from monocytes and support defining these cells as CD11c[Hi] MDC [13, 21, 27].

## The transcriptional program of Mtb-infected cells is affected by both pro- and anti-inflammatory cytokines

Gene expression by YFP[pos] vs. YFP[neg] CD11c[Hi] MDC, or YFP[pos] vs. YFP[neg] RM, were compared and DRGs were identified based on an FDR <0.05 and a ±2-fold-change. We identified 231 DRGs among CD11c[Hi] MDC and 424 DRGs in RM. The expression of the DRGs in YFP[pos] vs. YFP[neg] AM, CD11c[Hi] MDC, and RM is presented graphically (Fig 5A, 5B and 5C). DRGs included cytokines (IL-1α, IL-1β), mediators of microbial immunity (NOS2), and regulators of lipid and eicosanoid synthesis (Ptgs2, Alox5, PPARγ, Fabp1, Abca1). Using the Ingenuity pathway analysis platform, LPS/MyD88, IFNγ, TNF, and IL-1β, were identified as activating upstream regulators that likely explain the transcriptional differences between YFP[pos] vs. YFP[neg] cells (Fig 5D). Importantly, most of these upstream regulators were products of activated CD4 or CD8 T cells (e.g., IFNγ, TNF, GM-CSF, and IL-2), or reflected innate microbial sensing (e.g., MyD88, IL-6, TNF, IL-1β), indicating that T cell recognition of Mtb-infected cells may be driving these gene signatures. In addition, there were signals of inhibitory cytokines including IL-4, IL-10 and TGFβ, which could modify the degree of inflammation and even impair anti-microbial immunity.

To perform pathway analysis, sets of DRGs with an FDR <0.05 were assembled. These sets contained 288, 464, and 624 genes for AM, CD11c[Hi] MDC, and RM, respectively. The top pathway predicted to be activated in infected AM by Ingenuity was "neuroinflammation," which includes NOS2, COX2, VCAM1, SLC1A2, IL-1β, among others (Fig 5E). Cholesterol biosynthesis and LXR/RXR pathways were predicted to be inhibited in infected AM. Although other molecules including NOS2 and IL-1α were highly expressed by YFP[pos] CD11c[Hi] MDC, the predicted regulated pathways differed from AM (Fig 5F). The neuro-inflammation pathway was predicted to be downregulated in infected CD11c[Hi] MDC, and the chief differences were driven by downregulation of CXCR1, TLR9, and FOS expression. Interestingly, the IFNγ signaling pathway was not significantly activated in CD11c[Hi] MDC, despite changes in the expression of interferon regulated genes (e.g., SPP1, NOS2, IL1A). These data could reflect IFNγ acting on both uninfected and infected CD11c[Hi] MDC or inhibition of IFNγ signaling in a subset of MDC. The complement system, pattern recognition receptor, and inhibition of matrix metalloproteinases pathways were predicted to be activated in Mtb-infected RM (Fig 5G). GSEA analysis showed that infected AM, CD11c[Hi] MDC, and RM had similar transcription responses, but the responses of CD11c[Hi] MDC and RM were more similar to each other (Fig 5H).

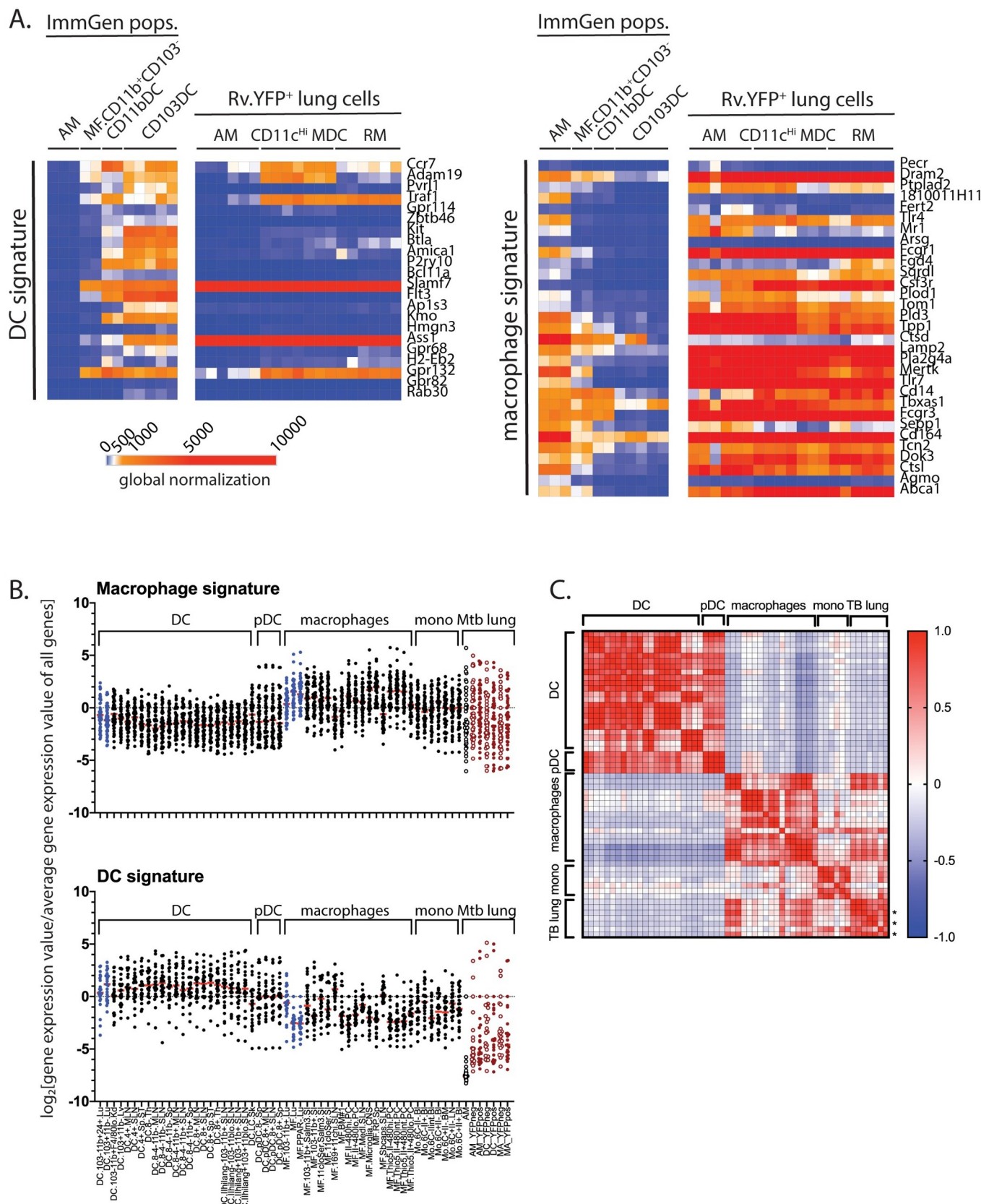

**Fig 4. The transcriptional signature of YFP^pos CD11c^Hi MDC is closely aligned with macrophages and monocytes.** A. Using well-described transcriptional gene signatures for DC [32] or macrophages [31], YFP^pos AM, CD11c^Hi MDC, and RM were compared to well-defined lung myeloid cell populations from uninfected mice (data from www.Immgen.org) using a heat map with global normalization. B. To quantify the macrophage and DC gene signatures, the log$_2$[specific gene expression / average total gene expression] was determined for 43 myeloid unique ImmGen myeloid cell populations and the seven populations we analyzed, grouped by cell type or "Mtb lung data", which refers to data derived from this study. Red bar, median. pDC, plasmacytoid DC. Blue, ImmGen lung populations; open black, AM from uninfected mice; open red; YFP^neg cells; closed red, YFP^pos cells. Not that the ImmGen data and our data obtained with different Affymetrix chips. C. Using a gene signature that distinguishes ImmGen DC from macrophages (see S4 Fig), a Pearson correlation analysis was performed for all the cell populations. "*" indicates YFP^pos populations.

## Identification of myeloid cells that are highly enriched for Mtb-infection

We identified a core set of 25 genes that was upregulated by all infected cell types >2-fold with an FDR <0.05 (Fig 6A). These genes were all upregulated in YFP^pos vs. YFP^neg AM, CD11c^Hi MDC, and RM (Fig 6B). Prominent among this gene set was NOS2, which is required for the synthesis of nitric oxide (NO), an antimicrobial molecule that also has anti-inflammatory effects [33, 34]. Two c-type lectins, Clec4d and Clec4e, were significantly upregulated in all three cell types. Clec4d (MCL) and Clec4e (Mincle) bind to Mtb ligands, (i.e., trehalose 6,6'-dimycolate, a.k.a cord factor) and are upregulated on myeloid cells after infection with other pathogens [35, 36]. To determine whether the corresponding proteins were regulated following Mtb infection, we used flow cytometry. We initially focused on CD38, CD14, ABCA1 and podoplanin, as their transcripts were highly upregulated by Mtb-infected AM, CD11c^Hi MDC, and RM compared to uninfected cells. As our gene expression studies were limited to non-PMN myeloid cell types, we excluded PMNs from our flow cytometric analysis.

In uninfected mice, fewer than 0.5% of lung non-PMN myeloid cells expressed CD38, CD14, and ABCA1 (Fig 7A, top row). In contrast, 3 weeks after infection, all three markers were present in various combinations on cells from the lungs of infected mice, and 10% of cells expressed high levels of all three markers (Fig 7A, bottom row, Q2, population "D"). Podoplanin was also enriched among Mtb-infected cells (S5 Fig). Importantly, most of the CD38^+CD14^+ABCA1^+ myeloid cells (hereafter referred to as 'triple-positive' or 'TP') were CD11c^Hi MDC (Fig 7B). Analysis of lung cells obtained from Rv.YFP-infected mice showed that CD38, CD14, and ABCA1 were more frequently co-expressed by Mtb-infected cells than by uninfected cells, as predicted by our transcriptional analysis (Fig 7C). Thus, 30–40% of the TP myeloid cells in the lungs of Mtb-infected mice were infected (i.e., YFP^pos), while only 1–2% of the non-TP cells were YFP^pos (Fig 7C).

We next systematically analyzed the expression of CD38, CD14, and ABCA1 among lung myeloid cells, after excluding PMNs, using a Boolean gating strategy. The four quadrants defined by CD38 and CD14 expression were analyzed for ABCA1, and the frequency of infected cells was determined for each of the resulting eight populations (i.e., A-H, see Fig 7A) (Fig 7D). The CD38^+/CD14^+ cells were the main cell population that expressed high levels of ABCA1 after infection (Fig 7A). The triple negative cell (population G (white bar), 'TN') were the largest subset accounting for ~55% of all myeloid cells; the other seven populations accounted for between 5–10% of myeloid cells (Fig 7D). These TN cells had a low rate of infection, especially in contrast to the TP cells (population D (red bar), of which nearly 40% were Rv.YFP^+ (Fig 7D). Thus, although they only account for ~10% of the lung myeloid cells, the TP cells account for ~60% of the non-PMN infected cells (Fig 7D). Furthermore, the MFI of Rv.YFP was significantly higher in the TP cells than in other the cell populations (Fig 7D). Since we have established that the MFI of Rv.YFP correlates with the number of bacteria/cell (Fig 2A and 2B), these data indicate that the TP cells have more bacteria per cell. Thus, CD38, CD14, and ABCA1 define a population of myeloid cells that harbor most of the bacteria in the lungs of infected mice.

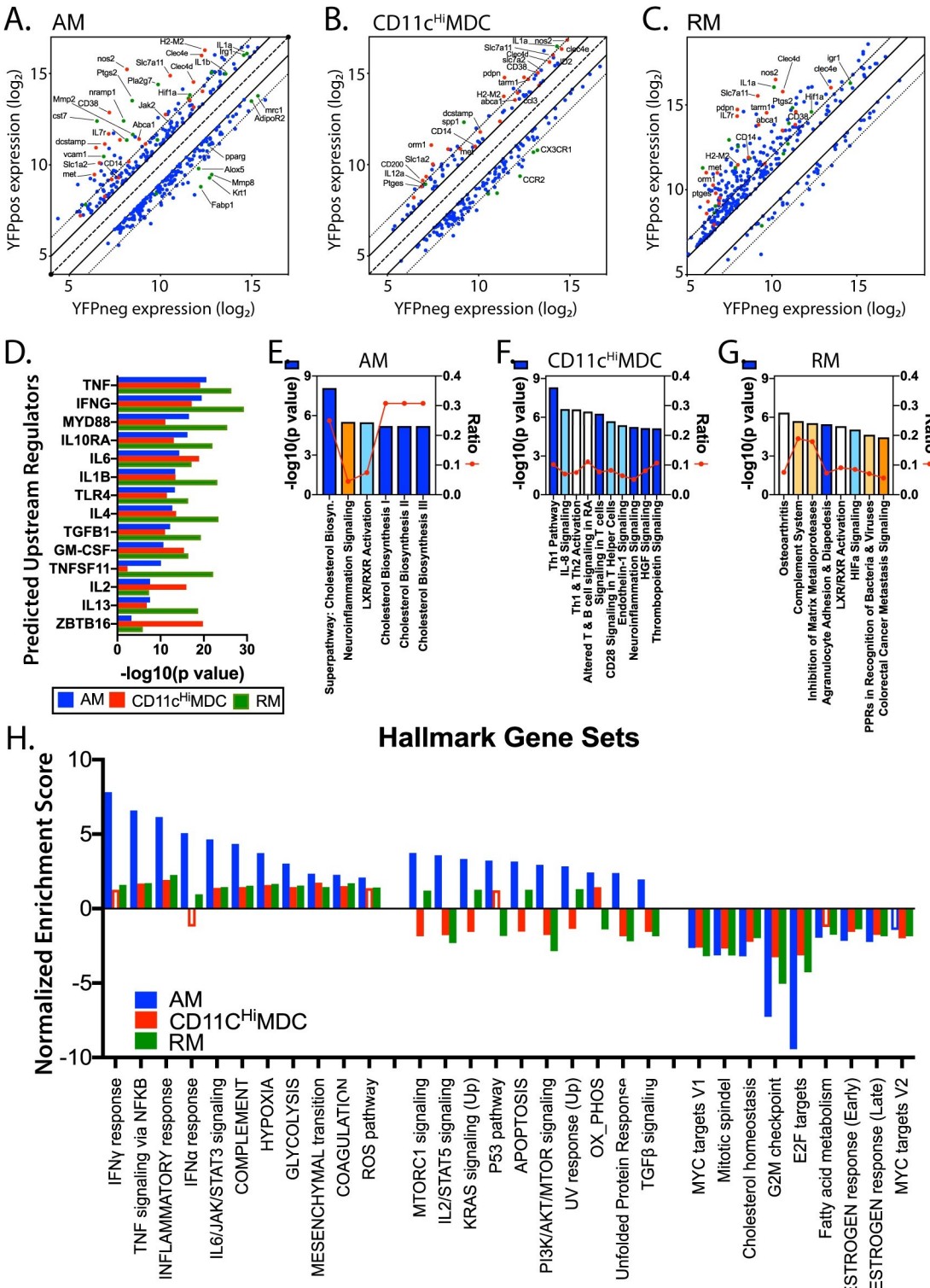

**Fig 5. Transcriptional analysis of infected AM, CD11c$^{Hi}$ MDC, and RM.** A, B, C. Using an FDR <0.05 and a fold-change >[2], we detected 370, 231, and 424 DRGs, between YFP$^{pos}$ and YFP$^{neg}$ cells in AM (A), CD11c$^{Hi}$MDC (B), and RM (C) from infected mice, respectively. For each of these cell types, the log$_2$ expression differences between YFP$^{pos}$ and YFP$^{neg}$ cells is plotted. The line of identity is the dashed line; the solid line represents ±2, and the dotted line is ±4-fold difference. Red dots identify genes that were differentially regulated in all three cell types. Green dots identify other genes of interest. The identities of some genes are annotated. D. Ingenuity pathway analysis (IPA) predicted upstream signals for each cell type based on the DRGs. Each bar

represents the $-\log_{10}$(p value). E, F, G. IPA predicted canonical pathways based on DRGs. Pathways predicted to be upregulated are shaded using warm colors; pathways predicted to be downregulated are shaded with cool colors. The bars indicate the $\log_{10}$ p values; the line indicates the fraction of genes from each pathway that were identified as DRGs. H. Top pathways predicted by GSEA for each cell type. Bars, normalized enrichment score (NES). FDR<0.05 (solid bars); FDR>0.05 (open bars).

## Infected 'TP' cells are activated

Mtb-infected AM, CD11c$^{Hi}$ MDC, and RM had an activated transcriptional signature consistent with upstream regulators IFNγ, TNF and TLR signaling (Figs 3, 5 and 6). The NOS2 transcript was highly upregulated in all three infected cell subsets (Figs 3, 5 and 6). We confirmed that NOS2 protein was induced in all three subsets (Fig 8A), roughly in proportion to their bacterial load (Fig 2D). To confirm that Mtb-infected cells expressed NOS2 protein, lung myeloid cells were analyzed after Rv.YFP infection. NOS2 was detected in 67% of Mtb-infected cells compared to only 4% of uninfected cells (Fig 8B). NOS2 has previously been shown to be expressed by Mtb-infected cells [9, 13, 37]. Similarly, intracellular pro-IL-1β protein was detected in 30% of YFP$^+$ cells in the lung (Fig 8B). Having found that infected cells are more likely to have a 'TP' phenotype, we assessed the phenotype of NOS2-expressing cells: 90% co-expressed CD14 and CD38 (Fig 8C). Conversely, CD38$^+$CD14$^+$ cells were highly enriched for NOS2 and IL-1β producing cells (Fig 8D). Finally, using Boolean gating to define eight populations of cells (i.e., populations A-H from Fig 7A), we find that TP cells had the highest frequency of NOS2 producers as well as the highest expression of NOS2 (Fig 8E). Thus, not only are lung myeloid cells expressing CD38, CD14, and ABCA1, more likely to be infected following Mtb infection, but they are also more likely to be expressing NOS2 and IL-1, which are associated with the anti-mycobacterial response.

## IFNγ drives TP cells to express CD38

We hypothesized TP cells are highly infected because they are more phagocytic than other subsets. To determine whether lung TP cells are more phagocytic, we obtained lung cells from mice infected with Rv.YFP 3 wpi, and measured their uptake of Rv.mCherry or yellow-orange (YO)-beads. As expected, Rv.YFP was preferentially found in the CD38$^+$CD14$^+$ cells (Fig 9A,

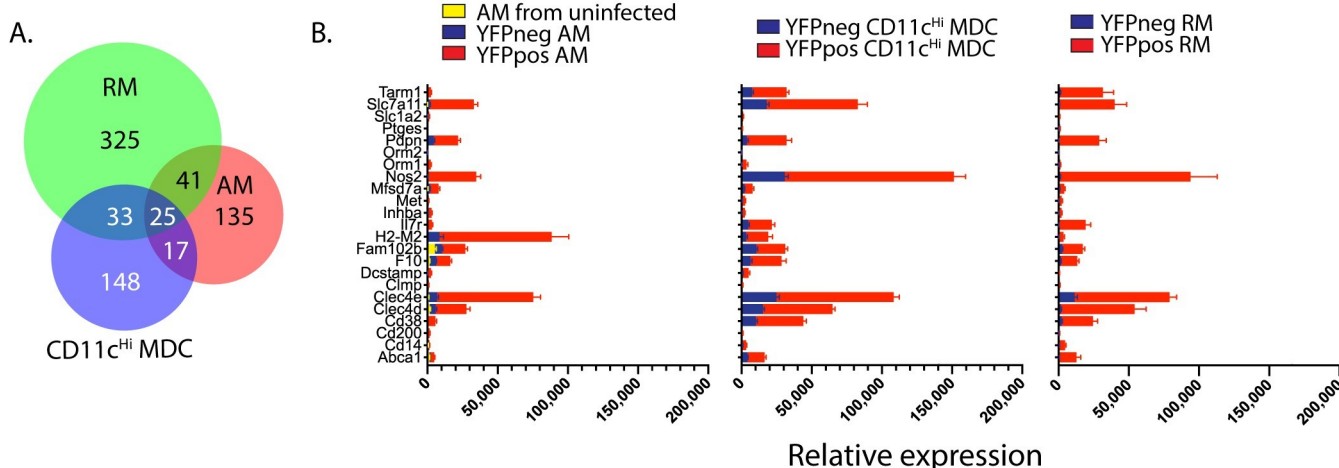

**Fig 6. 25 genes are upregulated genes in infected AM, CD11c$^{Hi}$ MDC, and RM.** A. The distribution of the 724 genes that are upregulated in YFPpos or YFPneg AM, YFPpos or YFPneg CD11c$^{Hi}$ MDC, or YFPpos or YFPneg RM. B. The expression levels of the 25 shared genes in YFPpos vs. YFPneg AM, CD11c$^{Hi}$ MDC, or RM. For AM, data from uninfected AM obtained from uninfected mice, is also shown.

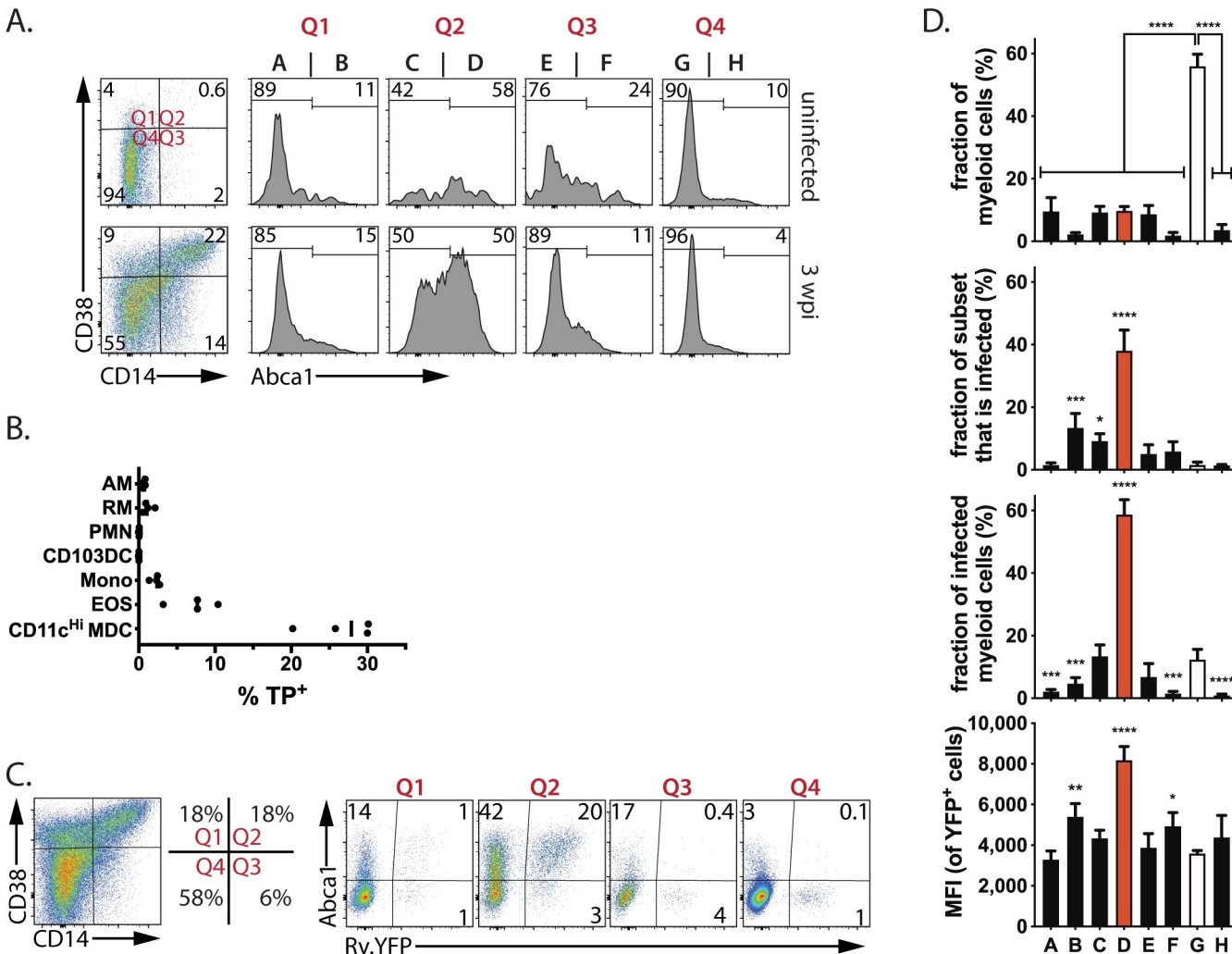

**Fig 7. CD38⁺/CD14⁺/ABCA1⁺ ('TP' cells) myeloid cells are highly Mtb-infected.** A. The expression of CD38, CD14, and ABCA1 by non-PMN myeloid cells from the lungs of B6 mice, 3 wpi. Quadrants are defined as follows: Q1, CD38⁺CD14⁻ᵛᵉ; Q2, CD38⁺CD14⁺; Q3, CD38⁻ᵛᵉCD14⁺; and Q4, CD38⁻ᵛᵉCD14⁻ᵛᵉ. Each quadrant was then further divided into Abca1 negative and positive cells. This Boolean gating based on CD38, CD14, and ABCA1 expression defines 8 populations (labelled 'A' through 'H'). B. The frequency of TP cells in each myeloid cell subsets. Each point represents one individual subject. C. CD38⁺CD14⁺ABCA1⁺ ('TP' cells) are enriched among YFP⁺ cells. Quadrants, based on CD38 and CD14 staining, were analyzed for YFP and ABCA1 expression. The highest frequency of YFP⁺ cells is found in CD38⁺CD14⁺ABCA1⁺ cells. D. For each of the gated cells populations labelled in Fig 7A as A through H, we determined: (1) the percentage of total myeloid cells in each gate; (2) the frequency of cells in each gate that are infected with Rv.YFP; (3) the contribution that each population makes to the total number of infected cells; and, (4) the MFI of the YFP⁺ cells in each gate. These data are representative of five independent experiments. Empty bars, 'TN' cells; red bars, 'TP' cells. Statistics, one-way ANOVA compared to 'TN' cells. Error bars, SD. *, p<0.05; **, <0.01, ***, <0.001, ****, <0.0001.

top row). By excluding the Rv.YFP infected cells, the phagocytic activity of the uninfected lung cells could be measured. Except for the CD38⁺CD14⁻ cells, which were poorly phagocytic, the YO-beads were also evenly distributed among cells that differed in their expression of CD14 and CD38 (Fig 9A, bottom row). The infection of myeloid cells in vitro by Rv.mCherry was independent of CD14 and CD38 expression (Fig 9A, middle row). These results indicate that a difference in phagocytosis cannot explain why TP cells are more infected than other myeloid cells.

Having identified that TP cells as frequently infected by Mtb, we asked whether host or microbial factors drive their generation. BMDC express CD14 and moderate amounts of

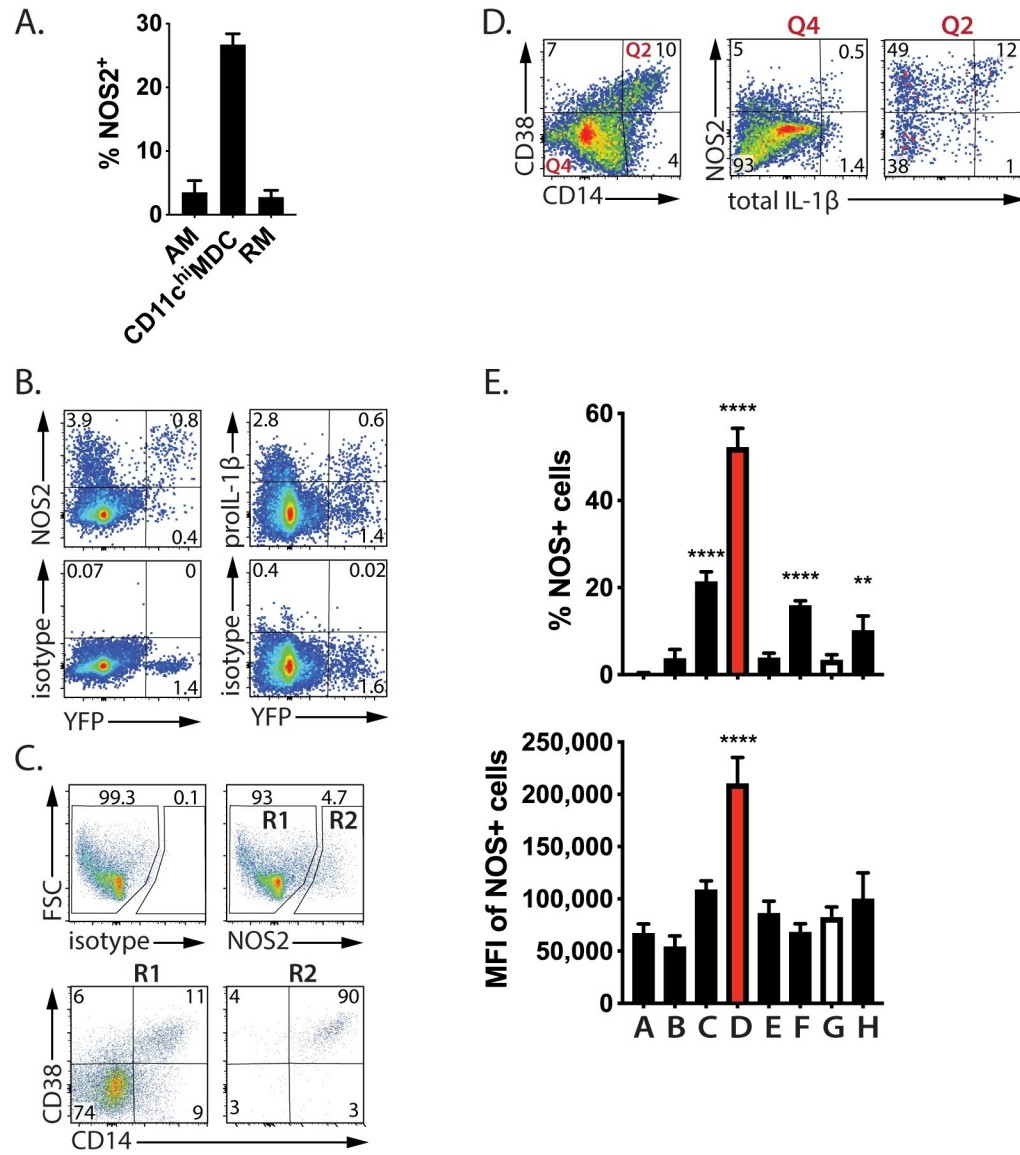

**Fig 8. TP cells are NOS2-producing.** A. The frequency of NOS2$^+$ cells among CD11c$^{Hi}$ MDC, AM and RM. B. Most Mtb-infected cells (Rv.YFP$^+$) produce NOS2 and many produce IL-1β. C. NOS2 was produced nearly exclusively by CD14/CD38 double positive cells. The expression of CD14 and CD38 by NOS2$^{-ve}$ (R1) and NOS2$^+$ (R2) cells was determined by flow cytometry. D. CD38$^+$/CD14$^+$ cells are highly enriched in cells producing NOS2 and IL-1β. CD14/CD38 double positive cells (Q2), or double negative cells (Q4) were analyzed for intracellular NOS2 and IL-1β expression. E. TP cells produce the highest amount of NOS2. For each of the cell populations expressing various combinations of CD38, CD14 and ABCA1 (A-H, defined by Boolean gating, see Fig 6), we determined the: (1) percentage of cells that produce NOS2; and, (2) the MFI of NOS2 expression. Empty bars, 'TN' cells; red bars, 'TP' cells. The plots were gated on non-PMN myeloid cells and are representative of three independent experiments. Statistics, one-way ANOVA compared to 'TN' cells. Error bars, SD. *, p<0.05; **, <0.01, ***, <0.001, ****, <0.0001.

CD38, while BMDM express CD14 but little CD38. A clear synergistic effect was observed when Mtb-infected cells were treated with IFNγ, an effect that was most pronounced for CD38 expression by BMDC and CD14 expression by BMDM (Fig 9B). In contrast to CD38 and CD14, ABCA1 levels were significantly reduced following IFNγ treatment, as previously described [38]. To determine whether IFNγ is required for the TP cell generation in vivo, we infected IFNγ KO mice. CD38 expression was lower, and fewer TP cells were detected

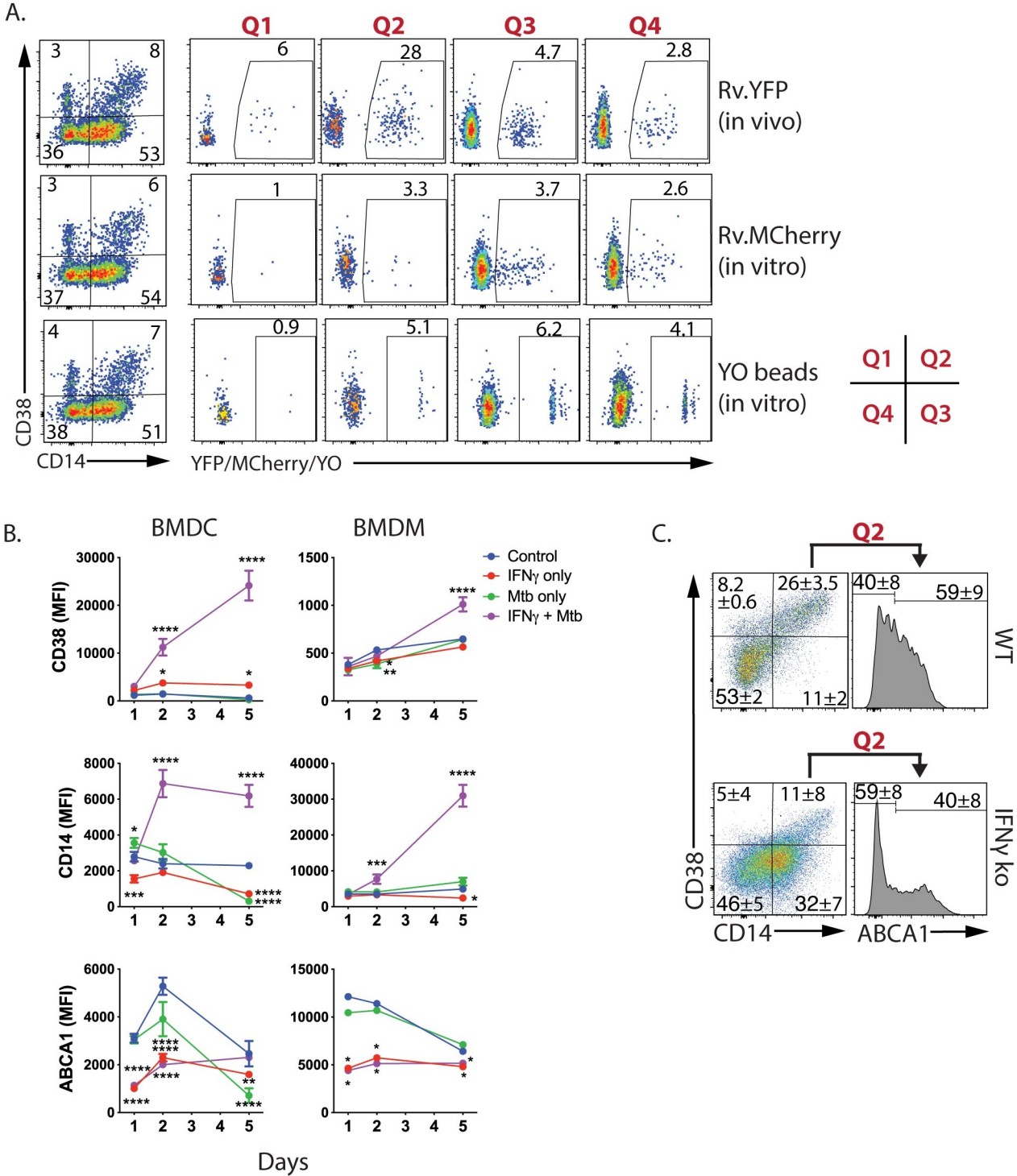

**Fig 9. TP cell phagocytosis and IFNγ responsiveness.** A. Lung TP myeloid cells are not more phagocytic or likely to become infected by Mtb. Four weeks after infection with Rv.YFP, single cell suspensions from the lung were obtained and infected in vitro with Rv.mCherry (MOI 5, 2.5 h). Cells were also incubated with YO-beads to measure phagocytosis. Top row, Distribution of Rv.YFP among CD14/CD38-expressing myeloid cells. After excluding the Rv.YFP+ cells by gating, the Rv.YFP-ve cells were analyzed for their uptake of Rv.mCherry (middle row), or phagocytosis of YO-beads (bottom row). B. IFNγ promotes CD38 and CD14 expression by myeloid cells. BMDC and BMDM were infected with Rv.YFP at an MOI of 0.2 in the presence or absence of IFNγ for 1, 2, or 5 days, and the expression of CD28, CD14, and ABCA1 were determined. C. IFNγ promotes TP cell generation in vivo. IFNγ KO or WT mice were infected with Rv.YFP and on day 25, non-PMN myeloid cells were analyzed for the expression of CD14, CD38, and Abca1. These data are from individually analyzed mice (WT, n = 3; IFNγ KO, n = 3).

compared to WT mice (Fig 9C). Thus, both in vitro and in vivo observations demonstrate that IFNγ drives the generation of TP cells that are highly infected.

## CD11c$^{Hi}$ MDC are highly infected by Mtb late during infection

We next addressed how TP cells evolve over time. Among non-PMN myeloid cells, there were few CD38$^+$CD14$^+$ cells two weeks after infection (Fig 10A). By 3 wpi, a discrete population of CD38$^+$CD14$^+$ cells was detected, and by 6 wpi, ~20% of non-PMN myeloid cells expressed CD38 and CD14 (Fig 10A). The kinetics of the development of CD38$^+$CD14$^+$ cells parallels the recruitment of T cells to the lung and is consistent with a role for IFNγ in the expression of CD38 and CD14 (Fig 9B and 9C). TP cells were located primarily in the lung parenchyma consistent with their high rate of infection (Fig 10B and 10C). Overtime, MerTK and CD64 expression increased on CD38$^+$CD14$^+$ cells and ultimately, most CD38$^+$CD14$^+$ myeloid cells expressed MerTK$^+$ and CD64, similar to the IM population defined by Huang et al [13]) (Fig 10A). Indeed, as IM increased in abundance in the lung between 3 and 6 wpi (Fig 1H, Fig 10A), the majority expressed CD38 and CD14 (Fig 10D). Importantly, it is this population (i.e., CD38$^+$CD14$^+$Mertk$^+$CD64$^+$) which is underlined{highly infected} (Fig 10E).

While the relative abundance of AM declined during infection, AM still represented 10% of all myeloid cells through four months after infection (Fig 10F). In contrast, CD11c$^{Hi}$ MDC rapidly increased between two and three weeks after infection and became the dominant mononuclear myeloid population by six weeks and continued to increase through four months post-infection. At three-, six-, and 16-weeks post infection, CD11c$^{Hi}$ MDC had the highest rate of infection, although as time passed, the overall frequency of Mtb-infected cells declines (Fig 10G, top row; S6 Fig). We next calculated the distribution of Rv.YFP$^+$ cells among the seven different myeloid cell types. After three weeks, most Mtb was detected in AM, monocytes, CD11c$^{Hi}$ MDC and PMNs. However, by six weeks, most Mtb was found in CD11c$^{Hi}$ MDC (Fig 10G, bottom row). Interestingly, not only did AM and CD11c$^{Hi}$MDC express high levels of class II MHC, but Mtb-infected AM and CD11c$^{Hi}$MDC, expressed similarly high cell surface class II MHC (Fig 11; S6 Fig). In contrast, only RM had heterogenous class II MHC expression. Thus, although there was no evidence that Mtb inhibited class II MHC expression, it remains to be determined whether infected cells expressing low levels of class II MHC could impair CD4 T cell recognition and control of intracellular infection. Considering the prominent infection rate of TP cells compared to non-TP cells, these data show that parenchymal TP CD11c$^{Hi}$ MDC cells are important cellular location for Mtb in the mouse lung.

## Discussion

The lack of definitive lineage markers that can identify discrete cell types during inflammation makes it difficult to determine whether certain types of myeloid cells ineffectively control bacterial replication and develop into a cellular niche where Mtb evades adaptive immunity. While Mtb initially infects AM, different cell types become infected as the infection progresses, and it isn't always clear whether differences in cell surface markers identify unique cell types or distinct activation states [10–12, 21, 28]. Improvements in multiparametric flow cytometry, the availability of fixation-stable fluorochromes and recombinant mycobacterial strains expressing fluorescent proteins have improved the capacity to study Mtb-infected cells *ex vivo* [9, 13, 22, 29]. We used an antibody panel which took advantage of SiglecF, a marker specific for AM and eosinophils, and our dump channel excluded lymphocytes, which express "myeloid markers" when activated, including CD11b and CD11c. We determined that YFP MFI could differentiate infected from uninfected cells and correlated with the MOI of Rv.YFP. Combining these tools allowed us to identify and sort YFP$^{pos}$ vs. YFP$^{neg}$ AM, CD11c$^{Hi}$ MDC,

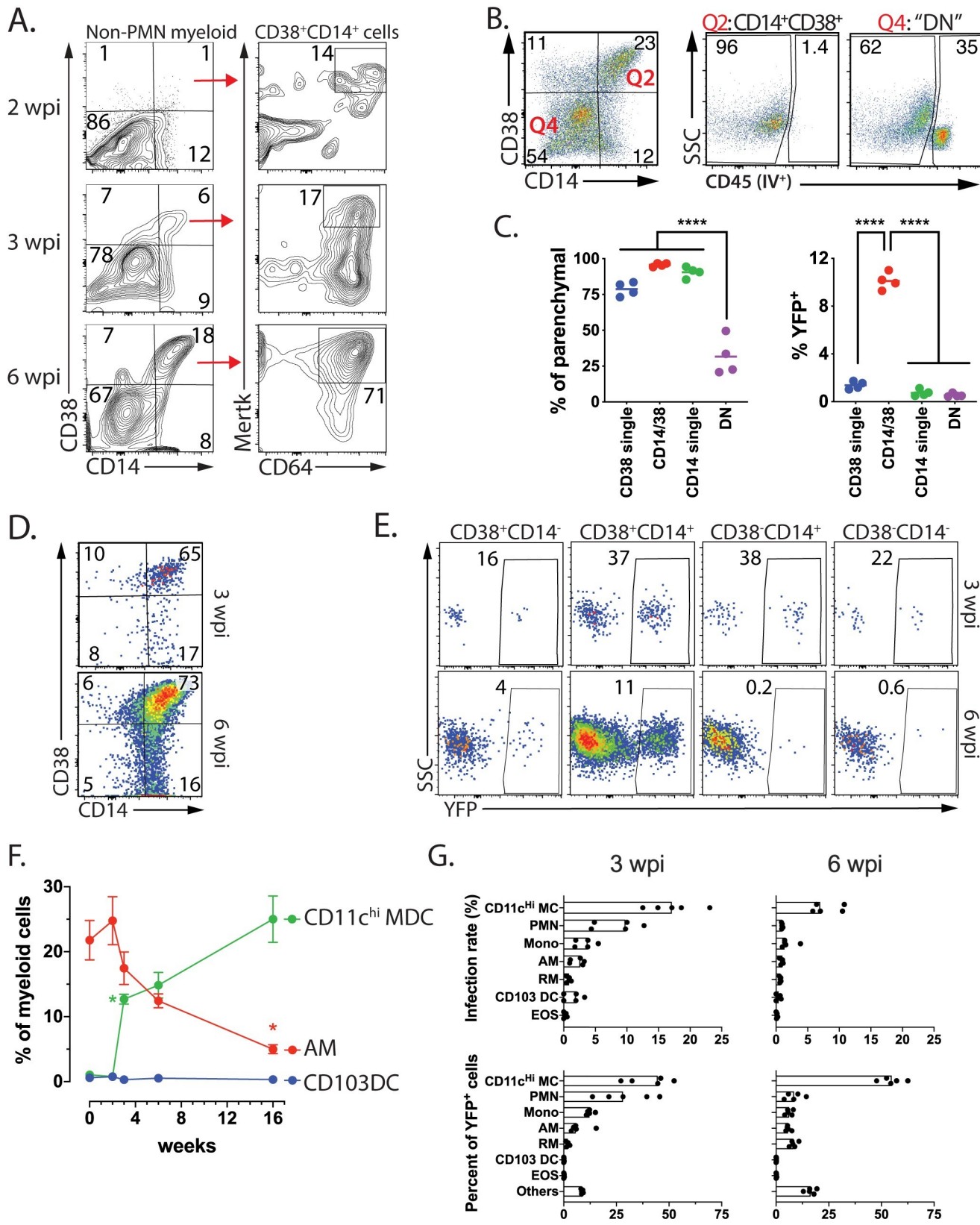

**Fig 10. TP CD11c^Hi MDC are frequently infected by Mtb.** A. CD64 and MerTK expression by CD14+/CD38+ myeloid cells at 2, 3, or 6 weeks after Mtb infection. B. CD14+/CD38+ myeloid cells (i.e., "Q2") are exclusively located in the lung parenchymal compartment while 35% of the CD14-/CD38- myeloid cells (i.e., "Q4") are in the vasculature staining from intravenously administered anti-CD45 mAb. C. Quantification of parenchymal location of CD14+/CD38+ myeloid cells CD38 staining (from Fig 10B) (left) and the frequency of YFP+ cells among the four subsets (right). D. A majority of CD64+/MerTK+ myeloid "IM" cells express CD14 and CD38, 3 and 6 wpi. E. The frequency of YFP+ cells among CD64+/MerTK+ myeloid "IM" cells based on their expression of CD14 and CD38 expression at 3 and 6 wpi. F. The frequencies of CD11c^Hi MDC, AM and CD103 DC among myeloid cells after infection with Rv.YFP over time. Error bars, SD. *, p<0.05; t-test compared to the prior timepoint. Each time point represents five mice. G. The infection rate (% YFP+) for each of the seven myeloid cell types (top row) and the contribution of each cell type to the total YFP+ population (bottom row). Non-PMN myeloid cells (A-E) or total myeloid cells (F, G) were analyzed. Each time experimental group consists of five mice and is representative of at least two independent experiments.

and RM from the lungs of infected mice. By comparing uninfected and infected cells within the same infected animal provided data on how these cells respond to infection.

Previously, Rothchild et al showed that the early innate AM response to Mtb (i.e., ≤10 days after infection) is characterized by a NRF2-driven response and is relatively anti-inflammatory [22]). This early phase of infection is notable for its lack of an inflammatory response and a delay T cell priming. In contrast, we find that infected AM are can be characterized by a pro-inflammatory state three weeks post-infection, during which time a robust Th1 response has been recruited to the lung. The infected AM population has a gene expression signature consistent with both an infected state (e.g., TLR signaling) and recognition by T cells (e.g., IFNγ signaling). Using a high dose intranasal Mtb model, Huang et al find that recruited macrophages (i.e., IM) are better at controlling Mtb growth than AM [13]. Our results after low-dose aerosol Mtb infection overlap considerably with their results. However, comparison of infected AM to uninfected AM from mice three to six weeks after infection, shows that Mtb-infected

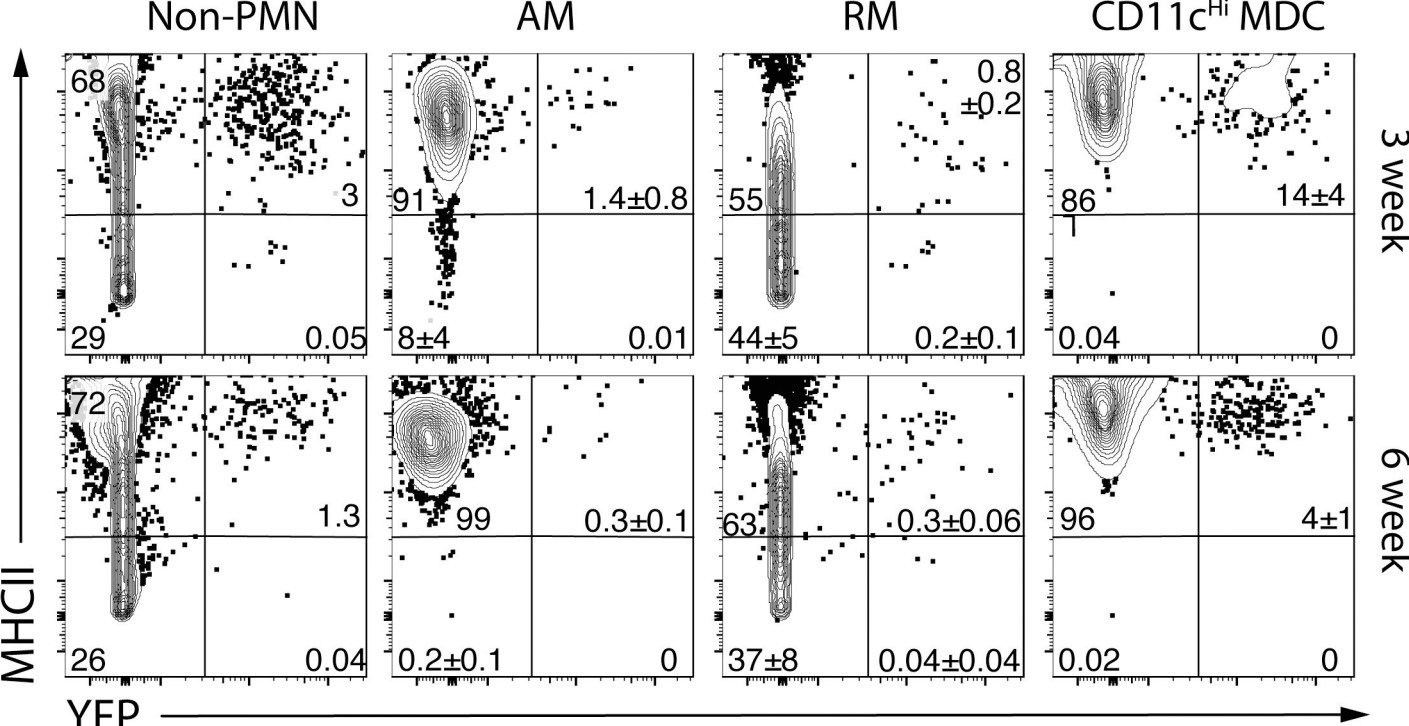

**Fig 11. Mtb-infected CD11c^Hi MDC have high class II MHC expression.** Class II MHC expression by uninfected and Mtb-infected (i.e., YFP+) non-PMN myeloid cells, or the AM, RM, and CD11c^Hi MDC subsets from the lungs of C57BL/6 mice, three and six weeks after infection by aerosol route with ~100 Rv.YFP. Numbers represent the quadrant means ± SD (n = 4–5 mice/time point).

AM are highly activated, upregulated IFNγ-regulated genes (e.g., NOS2, H-2, CD274 and CD38), upregulated IL-1 and COX-2, and downregulate Alox5 and PPARγ, compared to uninfected AM, all of which should foster an environment that restricts bacillary growth [34, 39–41]. An important caveat is our inability to discern whether the intracellular Mtb were alive or dead, as it is possible that T cells interact with macrophages containing dead bacteria. Nevertheless, the pulmonary T cell response peaks 3–4 weeks after infection, which is coincident with the deacceleration of bacterial growth, and the onset of immune control. Thus, T cell mediated control of infection correlates with the T cell activation of Mtb-infected cells.

We identified 25 genes that were significantly upregulated in Mtb-infected versus uninfected cells, in all three macrophage populations (i.e., AM, CD11c^Hi MDC, and RM). Among these, we validated the expression of CD14, CD38, ABCA1, and podoplanin by flow cytometry. Co-expression of CD14, CD38, and ABCA1 identified a population of highly infected lung myeloid cells. While CD14 is the receptor for LPS, it is also mediates uptake of Mtb and apoptotic cells [42, 43]. CD38 is an ectoenzyme that generates cyclic ADP-ribose and regulates intracellular calcium [44, 45]. ABCA1 is a cholesterol efflux pump regulated by PPARγ, a transcription factor that profoundly affects Mtb survival [39, 46, 47]. Finally, podoplanin is associated with the phagocytic potential of macrophages [48]. Thus, these molecules are positioned to restrict bacillary growth, or be subverted by Mtb to promote its survival.

We entertained four hypotheses to explain the high rate of Mtb-infection in these "triple-positive" cells. One was that CD14, CD38, and ABCA1 expression identify macrophages that are prone to infection. However, our *ex vivo* phagocytosis assay showed that TP cells from the lungs of infected mice were not preferentially infected. A second hypothesis was that these molecules were induced by intracellular Mtb. However, a large fraction of TP cells remained uninfected, excluding this possibility. Third, we considered whether the high infection rate of TP cells was secondary to the propensity to migrate into the parenchyma [9]. However, CD14 and CD38 single positive cells were also in the parenchyma; thus, location cannot account for the high infection rate of CD14^+CD38^+ cells. Our fourth hypothesis was that T cell recognition of infected cells drives macrophage activation and the expression of these surface markers. Indeed, the kinetics of T cell recruitment to the lung correlates with the appearance of TP cells, and IFNγ potentiates CD38 and CD14 induction. While ABCA1 expression can be inhibited by IFNγ [49], its regulation in vivo is likely to be more complicated as its level is affected by the local accumulations of cholesterol or other lipids. This hypothesis is supported by the IFNγ and TNF-driven gene signature of infected cells and is consistent with IFNγ's role in promoting CD38 expression, a recognized M1 marker [50].

Classifying different macrophages during inflammatory states is challenging because they are recruited, differentiate and become activated. We developed an antibody panel and gating schemes to distinguish distinct populations of myeloid cells based on our current understanding of myeloid cell ontogeny, which provides a clear framework for their classification [5]. This approach divides lung myeloid cells into tissue resident cells: AM (embryonic origin), CD103DC and CD11bDC (i.e., cDC1 and cDC2, from DC precursors) [51]. Other myeloid cells recruited to the lung, particularly during inflammation, are largely derived from monocytes. At steady-state, 2–3 distinct populations of interstitial macrophages (IM) have been described in an infected lung tissue [1, 2, 52]. These IM express CD64 and MerTK and are distinguished from AM by their higher expression of CD11b and MHCII, lower expression of CD11c, and absence of SiglecF. Monocyte-derived macrophages appear in the lung during inflammation such as tuberculosis [27, 53]. Mtb initially infects AM, which translocate into the lung interstitium [9, 22]. Following bacterial replication and AM cell death, the dispersed bacteria infect other cells. Three weeks after infection, most lung macrophages (i.e., MerTK^+CD64^+) are AM accompanied by few IM (Fig 1H). We find that most intracellular

Mtb was in cells that previously referred to as CD11bDC [27], but based on new data we now believe these cells are recruited monocytes (e.g., Ly6c$^{Hi}$CX3CR1$^{Hi}$MerTK$^{-ve}$) that differentiate into macrophages. Although we cannot rule out that some "true" CD11bDC are among the Mtb-infected cells, their overall gene expression pattern was similar to monocyte-derived macrophages (MDM, MoMa) [54], which others call IM [13], monocyte-derived DC (MoDC) [55], or MDC [32]. Eventually, the major cell population infected by Mtb is CD11b$^+$CD11c$^{hi}$ MHCII$^{hi}$MerTK$^+$Ly6C$^{lo}$ MDC that express CD14, CD38, and ABCA1. In addition, we detect a population of CD11b$^+$CD11c$^{lo}$MerTK$^-$Ly6C$^{lo}$ myeloid cells that we call RM. In uninfected mice, most cells with an RM phenotype, are in the vasculature, and could be Ly6C$^{lo}$ monocytes [27]. However, nearly all of the YFP$^+$ RM in the parenchyma are distinct from CD11c$^{Hi}$ MDC, based on their cytometric phenotype and transcriptional profile. These cells represent a minor fraction of the total Mtb infected cells in the lung.

The degree to which our murine model informs us about human disease is an important consideration. Macrophages with a cell surface phenotype of CD11c$^{Hi}$CCR2$^+$CX3CR1$^+$ are found in the intestinal wall of subjects with inflammatory bowel disease and are thought to be derived from CD14$^+$ monocytes [56, 57]. A limited number of studies have described the diversity of human myeloid cells during lung inflammation [58, 59], defined by their expression of CD206, CD11b, CD11c, CD14, and CD64. In macaque and human granulomas, proinflammatory macrophage expressing CD163, CD68, HAM56, and iNOS are observed predominantly in the inner region of the lesions granulomas [60]. It remains to be determined if these macrophages are the human equivalent of murine CD11c$^{Hi}$MDC.

The basis for restrictive versus permissive granulomas may be linked to how T cells recognize and inhibit intracellular bacterial growth, which could vary by the local lung microenvironment, the cell type infected, and the ability of the bacilli to evade detection [61, 62]. Increasingly, AM are regarded as a protected niche occupied by Mtb, as host resistance is enhanced following AM depletion in the mouse model [13, 14]. However, AM were depleted at an early stage of infection (2 wpi) in those studies, before the full impact of T cell immunity is manifest in the lung. Whether the difference between the ability of AM and IM to restrict bacterial growth is an intrinsic difference, and whether it can be modified through interactions with T cells will require additional studies. It is striking that in our study, two opposing forces are at work. On the one hand, immunity leads to a diminishing rate of infection, particularly among CD11c$^{Hi}$ MDC, which falls from ~20% at 3 wpi, to 3% by 16 wpi. On the other hand, CD11c$^{Hi}$ MDC, which express CD14, CD38, and ABCA1, become highly infected overtime, and nearly 75% of Mtb is found in these cells. Why do these cells develop into a major reservoir of Mtb is not clear; nor whether they are permissive or restrictive. Despite the high expression of MHCII molecules by CD11c$^{Hi}$ MDC, these cells might be poor APC [63]. We and others find that T cells are not efficient at recognizing macrophages containing few bacteria, and this could be an important mechanism of immune evasion in vivo [64, 65]. We expect that by defining the CD38$^+$CD14$^+$ABCA1$^+$ CD11c$^{Hi}$ MDC as a cell type that is abundantly infected both early and late after Mtb infection, we gain insight into how Mtb persists in vivo despite a vigorous anti-bacterial immune response and make possible strategies designed to target Mtb-infected macrophages.

## Methods

### Ethics statement

Studies involving animals were conducted following relevant guidelines and regulations, and the studies were approved by the Institutional Animal Care and Use Committee at the University of Massachusetts Medical School (Animal Welfare A3306-01), using the recommendations

from the Guide for the Care and Use of Laboratory Animals of the National Institutes of Health and the Office of Laboratory Animal Welfare.

## Mice

C57BL/6J, Rag-1-deficient (B6.129S7-Rag1tm1Mom), IFN-gamma KO (B6.129S7-Ifng<tm1 Ts>/J), MyD88 KO [B6.129P2(SJL)MyD88] mice were purchased from Jackson Laboratories (Bar Harbor, ME). All animal protocols used herein were approved by University of Massachusetts Medical School Animal Care and Use Committee.

## Generation of BMDCs and BMDM

Bone marrow cells were harvested from wild type C57BL/6 or MyD88 KO mice and were differentiated into BMDC and BMDM according to the protocols previously reported [21, 66]. BMDCs were prepared by culturing bone marrow cells in RPMI1640 medium (Gibco, Gaithersburg, MD) 10% fetal bovine serum (FBS) (Sigma), and 50 μM 2-mercaptoethanol, 10 mM HEPES, 2 mM L-Glutamine (complete media), supplemented with 10 ng/ml granulocyte-macrophage colony-stimulating factor (GM-CSF, R&D Systems, MN) for 8 days. Fresh media was added on days 2 and 4 to maximize BMDC yield. Non-adherent DCs were collected from 100 mm petri dishes by pipetting, counted, and plated in desired culture plates. BMDM were also generated from wild type C57BL/6 or MyD88 KO mice. BMDMs were prepared by culturing in complete media supplemented with 20% L929 conditioned medium for 8 days. Adherent cells were detached by using trypsin/EDTA solution (Gibco) for 10 min at 37˚C.

## Construction of YFP-expressing Mtb H37Rv

Bacteria expressing yellow fluorescent protein (sfYFP) were generated by transformation of H37Rv with plasmid PMV261, which constitutively expresses sfYFP under the control of the hsp60 promoter.

## In vitro infection

Rv.YFP was grown in BD Difco Middlebrook 7H9 (Thermo Fisher, Waltham, MA) supplemented with 10% OADC (Sigma-Aldrich, St. Louis, MO), 0.2% glycerol and 0.05% Tween-80 (both from Thermo Fisher) to an OD600 of 0.8–1.2, opsonized with TB coat (RPMI 1640, 1% heat-inactivated FBS, 2% human serum, 0.05% Tween-80), washed again and filtered through a 5-micron filter to remove bacterial clumps. The bacteria were counted using a Petroff-Hausser chamber and added to the culture at certain multiplicities of infection.

## In vivo infection and lung cell preparation

YFP-Rv was grown to an OD600 of 1.2 and was aliquoted to cryogenic vials, which were stored in a -80˚C freezer until use. A bacterial aliquot was thawed, sonicated in a cup-horn sonicator (Branson Ultrasonics Corporation, Danbury, CT) for 1 minute and then diluted in PBS-0.01% Tween-80 to 5 ml. The bacterial inoculum was previously titrated to deliver around 100 CFU to each animal. The inoculum was added to the nebulizer of a Glas-Col Inhalation Exposure System (Glas-Col LLC, Terre Haute, IN), and mice were exposed to infectious aerosol for 30 min. The number of Mtb deposited in the lungs was determined for each experiment, by plating undiluted lung homogenate from a subset of the infected mice within 24 hours of infection. The inoculum varied ranging between 37–90 CFU. To isolate total lung leukocytes, lungs were perfused by slowly injecting PBS into right ventricle immediately after mice were killed. The lungs were minced with a gentleMACS dissociator (Miltenyi) and digested (30 min, 37˚C) in

250 U/ml collagenase and 60 U/ml DNase (both from Sigma-Aldrich). Lung cell suspensions were passed through a 70-μm and 40-μm strainers sequentially to remove cell clumps. Lung cells were resuspended in autoMACS running buffer (Miltenyi) that contains BSA, EDTA, and 0.09% sodium azide for subsequent staining.

## Flow cytometry analysis

The Ab panel to define 7 myeloid subsets consisted of anti-CD45 (clone 30-F11) labeled with BV650, anti-Ly6G (clone 1A8) with APCfire 750, anti-CD11c (clone N418) with APC, anti-Siglec F (clone E50-2440) with PE, anti-CD11b (clone M1/70) with BV421, anti-Ly-6C (clone HK1.4) with PerCP-Cy5.5, anti-CD103 (clone M290) with BUV395,. The dump channel was used to exclude T cells, B cells, and NK cells for efficient myeloid cell analysis and included anti-Thy1.2 (clone 30H12), anti-CD19 (clone 6D5), and anti-NK1.1 (clone PK136) with PE-Cy7. The Ab panel to define myeloid subsets and TP cells included anti-CD45 (clone 30-F11) labeled with BV650, anti-CD38 (clone 90) with AF-647, anti-CD11b (clone M1/70) with AF594, anti-Abca1 (clone 5A1-1422) with PE, anti-Ly-6C (clone HK1.4) with PerCP-Cy5.5, anti-Siglec F (clone E50-2440) with APC-R700, anti-CD14 (clone Sa14-2) with APC-Cy7, anti-CD103 (clone M290) with BUV395, anti-CD11c (clone N418) with BV421. The dump channel for this Ab panel included anti-Thy1.2 (clone 30H12), anti-CD19 (clone 6D5), anti-NK1.1 (clone PK136), and anti-Ly6G (clone 1A8) with PE-Cy7. The Ab panel for Aurora to define iNOS and IL-1 beta in addition to myeloid subsets and TP cells includes anti-CD45 (clone 30-F11) labeled with BV650, anti-Ly6G (clone 1A8) with APCfire 750, anti-CD38 (clone 90) with Pacific Blue, anti-CD11b (clone M1/70) with BV421, anti-Abca1 (clone 5A1-1422) with PE, anti-CD11c (clone N418) with BV570, anti-Siglec F (clone E50-2440) with APC-R700, anti-CD14 (clone Sa2-8) with PerCP-eFluor710, anti-CD103 (clone 2E7) with BV711, anti-Ly-6C (clone HK1.4) with PerCP-Cy5.5, anti-CD11c (clone N418) with BV421, anti-iNOS (clone CXNFT) with PE-eFluor610, and anti-IL-1 beta (clone NJTEN3) with APC. NJTEN3 recognizes pro-form of IL-1b. The dump channel for this Ab panel included anti-Thy1.2 (clone 30H12), anti-CD19 (clone 6D5), and anti-NK1.1 (clone PK136). Abs were purchased from either Biolegend, BD, or eBiosciences. Live/dead viability staining was done using Zombi Aqua (Biolegend).

Freshly prepared lung cells were incubated with 5 ug/ml of anti-mouse CD16/32 (clone 2.4G2, BioXCell, Lebanon, NH) for 10 min at room temperature to block non-specific binding through Fc receptors. Then, the same volume of Ab cocktail was added, and cells were incubated with Abs for 30 min at 4°C. Staining for Fc block and surface markers was done in autoMACS running buffer. Cells were then washed with PBS and stained with Zombi Aqua diluted in PBS. For intracellular staining of iNOS and IL-1 beta, lung cells were stained ex vivo without in vitro stimulation. The lung cells were first stained for cell surface markers and live/dead viability, and then fixed and permeabilized using Perm/Fix reagents (BD) according to the manufacturer's recommendation. For ICS staining, cells were incubated in perm buffer (BD) containing an antibody cocktail diluted for 20 min at 4°C.To inactivate the bacteria, samples were fixed with 1% paraformaldehyde/PBS for 1 hour at room temperature and then washed with MACS buffer. FlowJo Software (Tree Star, Portland, OR) was used to analyze the data. Single cells were gated by forward scatter area versus height and dead cells were excluded by live/dead staining.

## Cell sorting

Cells were sorted using BD FACSAria IIU Cell Sorter located in the biosafety level 3 lab in University of Massachusetts Medical School. WT C57BL/6 mice were infected with YFP-Rv

intratracheally at an inoculum of 5000. This high dose of YFP-Rv was intended to obtain suffi-cient numbers of infected cells sorted for Genechip analysis. 3 weeks later, Lung cell suspen-sions were obtained and stained for surface markers, so that YFP- and YFP+ cells from each AM, CD11c^Hi MDC, and RM were sorted. Because the maximum number of cell separation is four with the sorter, only two subsets of myeloid cells were collected at each experiment. The Ab panel for sorting AM and CD11b DC includes anti-CD45 (clone 30-F11) labeled with BV650, anti-CD11c (clone N418) with PerCP-Cy5.5, anti-CD11b (clone M1/70) with BV421, anti-CD103 (clone 2E7) with PE-Cy7, anti-Siglec F (clone E50-2440) with APC-R700. The Ab panel for sorting CD11c^Hi MDC and RM includes anti-CD45 (clone 30-F11) labeled with BV650, anti-CD11c (clone N418) with APC, anti-CD11b (clone M1/70) with BV421, anti-CD103 (clone 2E7) with PerCP-Cy5.5, anti-Siglec F (clone E50-2440) with PE-Cy7, anti-Ly6C (clone HK1.4) with AF700, Ly6G (clone 1A8) with APC-Cy7. Cells were stained with Zombi Aqua to exclude dead cells. Target cells were separated 4-way through 85 um nozzles into poly-propylene FACS tubes containing 2 ml of FBS.

## GeneChip analysis

Cells were obtained from 3–5 individual mice for each cell type, from two independent experi-ments. Sorted cells were subjected to RNA isolation by using Trizol according to the manufac-turer's protocol. The isolated RNA was used to generate cRNA which was then biotinylated and prepared according to the Affymetrix GeneChip 3′ IVT Express Protocol from 150 ng of total RNA. Following fragmentation, 10 µg of cRNA was hybridized for 16 h at 45˚C on Mouse Clariom D arrays (Thermo Fisher). Arrays were washed and stained in the Affymetrix Fluidics Stations 450. The arrays were scanned using an Affymetrix GeneChip Scanner 3000 7G. Initial QC was performed with Transcription Analysis Console 3 using the RMA algo-rithm with quantile normalization and general background correction. Samples that failed QC were excluded from further analysis. Genes differentially expressed between *YFP- and YFP + cells within each subset* were determined at a false discovery rate (FDR) of 0.05, a fold change ≥2, and an absolute expression of >120. Heat maps were made using Morpheus (https://software.broadinstitute.org/morpheus) using hierarchical clustering by rows and row normali-zation. Venn diagrams were made using BioVenn [67]. Gene enrichment scores were deter-mined using the Gene Set Enrichment Analysis (GSEA) program [68, 69], along with the MSigDB v6.2 gene signature database. Inguenity was used to predict the upstream regulators and predict the most affected canonical cellular pathways. As described above, genes differen-tially expressed between YFP+ and YFPneg cell populations (FC≥2; absolute expression >120) were used as input data and the results are reported as $\log_{10}$(p value) and in the case of the canonical pathway prediction, whether the pathway is predicted to be activated or inhibited.

## Supporting information

**S1 Fig. Identification of 7 populations of myeloid cells in the lungs of Mtb-infected mice.** CD11b and CD11c are frequently used to distinguish alveolar macrophages (AM) from pul-monary dendritic cells (DC): AM are generally CD11b^lo CD11c^hi MHCII^low-int, while DC are CD11b^int-hi CD11c^hi MHCII^hi. This strategy works well for uninfected mice housed under con-ventional conditions. However, during inflammation, AM upregulate both CD11c and MHCII. Therefore, the following scheme takes advantage of SiglecF to identify AM, independ-ently of MHCII. Single cell suspensions obtained from the lungs of mice were gated based on (a) scatter, (b) singlets, and then dead cells excluded (c). Leukocytes in the lung were identified based on CD45 staining (d) and lymphoid cells were excluded by using a dump channel that

consisted of anti-Thy1, CD19, and NK1.1 (e). Neutrophils (PMNs, "1") were identified based on Ly6g expression (f). AM ("2") were identified by their siglecF$^{Hi}$CD11c$^{Hi}$ phenotype, while eosinophils (Eos, "3") expressed lower amounts of SiglecF and no CD11c (i.e., siglecF$^{int}$CD11c$^{-}$) (g). CD11c$^{int}$CD11b$^{Hi}$ cells were identified as recruited macrophages (RM, "5") and monocytes, and monocytes were further distinguished by their Ly6C expression ("4"). Finally, the remaining CD11c$^{Hi}$ cells were designated as DC, and 2 different subsets identified by their expression of CD103 (i.e., CD103DC or cDC1, "6") or CD11b (i.e., CD11bDC or cDC2, "7"). In the Mtb infected lung, siglecF$^{-}$CD11c$^{Hi}$CD11b$^{Hi}$ cells are likely to be a mixture of CD11bDC and CD11c$^{Hi}$ MDC.
(PDF)

**S2 Fig. A "Dump" strategy excludes lymphoid cells.** In addition to eliminating debris, doublets, and dead cells from our analysis, an essential part of our strategy was to exclude lymphoid cells. After gating on CD45$^{+}$ cells, lymphoid cells were identified by using a "dump" gate consisting of a mixture of antibodies specific for Thy1, CD19, and NK1.1 (a). Although these markers primarily identify lymphoid cells, some lymphoid cells are known to express low levels of CD11b, CD11c, and SiglecF, particularly when activated, which could lead to misclassification of cell types. Although there are few Dump+ cells that express CD11b, CD11c, or SiglecF in the lungs of uninfected mice (b, c), one sees a more complicated pattern of CD11b and CD11c expression by CD45$^{+}$ cells by cells isolated from the lungs of Mtb-infected mice (S2b, left), by CD45$^{+}$dump$^{-}$ cells (center), or by lymphoid cells (right). This strategy is even important when CD11b, CD11c, and SiglecF staining are used to identify lung myeloid cells (c). Thus, our staining panel and gating strategy identify seven major populations of distinct myeloid cells that are present in the lungs, independently of *M. tuberculosis* infection. This strategy facilitates accurate enumeration of myeloid cells and measurement of phenotypic changes during infection, independent of the markers used to identify the different cell populations. A possible confounder is whether activated myeloid cells express Thy1, CD19, or NK1.1 lineage markers. Analysis of Mtb-infected RAG knockout mice, which lack B and T cells but still have NK cells, shows that few if any activated cells (i.e., MHCII$^{+}$) in the AM or MDC gate express the "dump markers" (d). As for the RM population, there were "Dump+" cells, which were presumably NK cells, but none in the MHCII$^{+}$ region. The cells show up in the RM gate as they lack expression of the other markers used to identify distinct myeloid populations (i.e., they lack SiglecF, CD11c, Ly6c, and MHCII). Additionally, we determined the percentage of cells in the dump channel that were YFP+ (e). Cells in the myeloid gate were ~30 times more likely to be infected than cells in the dump gate (~4.2% vs 0.14%). Thus, the dump channel excludes very few infected cells from subsequent analysis.
(PDF)

**S3 Fig. Identification of AM, CD103DC, and CD11bDC during Mtb infection.** CD11b and CD11c can be used to discriminate AM and DC in uninfected mice; however, they need to be used in conjunction with other markers during Mtb infection. For example, in uninfected mice, CD11b$^{lo}$CD11c$^{hi}$ cells are mostly AM (S3A Fig, left panel). Further analysis of the R1 gate shows that most of the cells are AM with some CD103DC (S3B Fig, top left panel). Similarly, CD11b$^{hi}$CD11c$^{hi}$ cells (S3A Fig, R2 gate) are nearly all CD11bDC (S3B Fig, bottom left panel). However, during Mtb infection, the pattern is more complex. The few CD11b$^{lo}$CD11c$^{hi}$ cells (S3A Fig, R1 right panel) are a mixture of AM and DC (S3B Fig, R1), and the CD11b$^{hi}$ CD11c$^{hi}$ cells are mostly CD11bDC and MDC but also AM (S3B Fig, R2). In contrast, using the gating scheme described above (S1 Fig), AM and eosinophils are unambiguously identified based on siglecF and CD11c, independently of infection (S3C Fig). The CD11c$^{hi}$siglecF$^{lo}$ cells define DC (Fig 3C, "DC gate"), while the CD11c$^{low-interm}$siglecF$^{lo}$ cells are RM and monocytes.

The cells in the DC gate include both CD103DC and DC11bDC before infection, but after infection, CD11bDC dominate. As we discuss later, we think that although there are CD11bDC in this population, most of the cells are CD11c$^{hi}$ monocyte-derived cells (CD11c$^{hi}$ MDC).
(PDF)

**S4 Fig. Gene expression analysis of YFP$^{+}$ "CD11bDC" from the lungs of Mtb-infected mice reveals that they closely resemble CD11c$^{Hi}$ MDC.** A. Using well-described transcriptional gene signatures for AM, macrophages, DC and monocytes [2, 32], YFP$^{pos}$ AM, CD11c$^{Hi}$ MDC, and RM were compared to well-defined lung myeloid cell populations from uninfected mice (data from www.Immgen.org) using a heat map with global normalization. The YFP$^{pos}$ AM have a transcriptional profile matching AM, while YFP$^{pos}$ CD11c$^{Hi}$ MDC and resemble the monocyte/macrophage lineage. B. We developed a 21 gene signature (here, referred to as "ImmGen 1") that distinguishes ImmGen DC from macrophages. An analysis using the expression of the 15 genes that are preferentially expressed in macrophages (top), the 6 genes that are preferentially expressed in DC (middle), or the combined 21 gene signature is quantified for 43 myeloid unique ImmGen myeloid cell populations, grouped by cell type. Red bar, median. pDC, plasmacytoid DC. Blue, ImmGen lung populations. C. The list of genes that make up the 21 gene signature.
(PDF)

**S5 Fig. Infected cells display disproportionately high expression of Podoplanin.** Lung cells were obtained from uninfected mice, or mice four weeks after aerosol infection with Rv.YFP. Podoplanin expression was analyzed by flow cytometry after gating on live CD45+ cells, excluding T cells, B cells, NK cells, and PMN. The black numbers are the percentage of events in each quadrant. The red numbers are the percentage of YFP$^{-ve}$ or YFP$^{+ve}$ events that express podoplanin.
(PDF)

**S6 Fig. Distribution of Mtb among different cell types and expression of MHC II by infected cells late during infection.** The H37Rv.YFP strain we use was transformed using a plasmid that expresses sfYFP driven by a constitutive promoter. These plasmids are lost over time and at late timepoints ($\geq$16 wpi) most infected cells no longer express the plasmid based on the lack of fluorescence of CFU grown from lung homogenates. Thus, we cannot rule out that there was preferential plasmid loss in certain cell types or differential degradation of YFP. However, myeloid cells could still be detected at late time points up till 9 months after infection. The distribution of YFP among different cell types (A) and their expression of MHC II (B) 4 months after infection follows a similar trend as observed at 3 and 6 wpi (see Figs 9 and 10). However, because of the issue of plasmid loss, one must interpret these data cautiously.
(PDF)

## Author Contributions

**Conceptualization:** Jinhee Lee, Samuel M. Behar.

**Formal analysis:** Jinhee Lee, Christina Baer, Samuel M. Behar.

**Funding acquisition:** Samuel M. Behar.

**Investigation:** Jinhee Lee, Shayla Boyce, Jennifer Powers, Christina Baer.

**Project administration:** Samuel M. Behar.

**Supervision:** Christopher M. Sassetti, Samuel M. Behar.

**Writing – original draft:** Jinhee Lee, Samuel M. Behar.

**Writing – review & editing:** Jinhee Lee, Shayla Boyce, Jennifer Powers, Christina Baer, Christopher M. Sassetti, Samuel M. Behar.

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
