## [Decision Letter · Decision Letter 0]

20 Mar 2020

Dear Dr. Behar,

Thank you very much for submitting your manuscript "A subset of CD11cHi monocyte-derived macrophages are the dominant niche for Mycobacterium tuberculosis" for consideration at PLOS Pathogens. As with all papers reviewed by the journal, your manuscript was reviewed by members of the editorial board and by several independent reviewers. In light of the reviews (below this email), we would like to invite the resubmission of a significantly-revised version that takes into account the reviewers' comments.

We cannot make any decision about publication until we have seen the revised manuscript and your response to the reviewers' comments. Your revised manuscript is also likely to be sent to reviewers for further evaluation.

Sincerely,

Marcel A. Behr

Associate Editor

PLOS Pathogens

Michael Wessels

Section Editor

PLOS Pathogens

Kasturi Haldar

Editor-in-Chief

PLOS Pathogens

orcid.org/0000-0001-5065-158X

Michael Malim

Editor-in-Chief

PLOS Pathogens

orcid.org/0000-0002-7699-2064

Reviewer's Responses to Questions

**Part I - Summary**

Reviewer #1: The manuscript by Lee and colleagues titled “a subset of CD11chi monocyte-derived macrophages are the dominant niche for Mycobacterium tuberculosis” addresses a rapidly evolving and highly significant area of tuberculosis research. As our appreciation of macrophage ontogeny and heterogeneity expands it is important to generate a functional appreciation of how the different macrophage subsets contribute to control or progression of tuberculosis in vivo. In this paper the authors describe a new subset of monocyte derived macrophages that are defined by the surface marker profile CD11b+CD11c+ MHCIIhi, and characterized by additional transcriptional profiling. When they examined the distribution of YFP-expressing Mtb in the different macrophage populations they found that these cells contained an unusually high burden of bacteria. On the basis of these observations they refer to these cells as the “dominant niche” for Mtb in the mouse lung (9 times in the text).

I find the data presented in the paper to be of high quality and they are undoubtedly of value to the field at this particular time. However, I think the repeated use of the term “dominant niche”, which implies functional significance beyond mere abundance, to be a considerable exaggeration of what has actually been demonstrated.

1. There is no information on the status of the bacteria in these cells, and they could be fully viable and replicative, or fully under host control.

2. There is no manipulation of this host cell population to demonstrate that they play any role of significance to either disease control or progression. To be able to claim that they are the “dominant niche” one would expect to see some experimental validation.

Absent any functional information the claim that this is the “dominant niche occupied by M tuberculosis long-term” is conjecture. As this remains an unproven hypothesis it should be presented as such and not sold as proven fact. The paper by Huang and colleagues included functional manipulation of cell subsets and demonstrated impact on bacterial burden, and more recently the same group published Dual RNA-seq analysis showing the “matching” stress signatures of pathogen and host cell subset (Pisu et al. Cell Reports 2020). I fully accept that these papers were generated with data from a single early time point and that the conclusions and findings in these papers do not refute the current data in this manuscript, which emphasize later time points in infection progression and probes the granularity of monocyte-derived cell subsets. However, the earlier papers included fitness reporter bacterial strains and experimental manipulation of the cell subsets to demonstrate their functional significance in vivo and the current study lack any comparable data.

The data in the current paper opens the possibility to the presence of a permissive subset of recruited monocyte-derived cells that may have considerable significance post development of an acquired immune response. I agree this is exciting but on the basis of the data presented it is a possibility and not a fact, and its significance remains to be established experimentally.

Reviewer #2: This manuscript by Lee et al examines the intracellular niche of M. tuberculosis (Mtb) during chronic infection. Although this topic has been studied extensively by several other groups, this work does advance new knowledge on the subject. That said, the paper could be written more clearly and in a more focused manner. I have several comments and questions which should be addressed.

Reviewer #3: The ability to distinguish between different cells of the myeloid lineage and their ontogeny particularly macrophages has been a major challenge in the context of Mtb infection. Isolating discrete populations from the heterogenous macrophage populations remains a barrier. In the current submitted manuscript; Lee et al., identify using flow cytometric analysis a subset of CD11chi monocyte derived cells which have an increased presence in the lung post Mtb infection. Moreover these cells seem to be induced or influenced by IFNγ as per their gene expression signature and evidence from the IFNγR KO mouse. In general the manuscript has been prepared thoughtfully and is well written, particularly the discussion. Some minor additions and alterations are needed to fully explain and add clarity to readers on the conclusions of the data presented. The data presented in the study thus far is quite comprehensive and well supported.

**Part II – Major Issues: Key Experiments Required for Acceptance**

Reviewer #1: If the authors wish to argue the functional significance of this cell subset as a niche that supports infection they need to include data demonstrating functional impact. However, I would be perfectly satisfied by a more circumspect handling of the interpretation of the data. I think the hypothesis is important, our knowledge of lung macrophages is woefully incomplete, and these data are timely and of value to the field.

Reviewer #2: 1) To me the most important finding of the paper is that the CD11c+CD11b+ myeloid population that is the predominant niche for Mtb during chronic infection, and has been referred to as dendritic cells in most of the TB literature for years are actually monocyte-derived cells that are most akin to macrophages (this is also the title of the manuscript). The best evidence for this would be the RNA-Seq analysis shown in Fig. S4, as the rest of the data does not conclusively make this point. Why is the most important figure shown only as a supplemental figure? The figure legend suggests this is old data that was performed using sorting “based on a limited number of cell surface markers.” If so, as this is the cornerstone of the manuscript, I believe this analysis should be repeated sorting cells using markers that they use throughout the manuscript and that they currently think is optimal. In addition, although the heat maps are helpful, is there a bioinformatic platform that could be utilized to better assess the relatedness between the various populations? I think this central message could be made more clearly and more rigorously. I think this analysis would be an important contribution to the field, but it needs to be done well and to be presented prominently in the paper.

2) , as shown by RNA-Seq (although this data is buried in Fig. S4). Although previous work by the authors and others has shown that at least some of these cells are derived from monocytes, several possibilities have remained: 1) that although some are monocyte derived, most are still conventional DCs that perhaps express some monocyte markers during the inflammatory milieu of TB, or 2) Most are monocyte-derived, but for all intensive purposes they differentiate into cells that are largely indistinguishable from DCs. Although it cannot be ruled out that a small subset of these cells may be true DCs, as the authors acknowledge, Fig S4 is the most convincing data to date that most of these are monocyte-derived cells that most resemble macrophages. This should be highlighted more and I believe this data should be moved to be one of the primary figures.

3) Instead of highlighting the above, the authors seem to spend excessive time (half of Figure 1 and again returning to in in Figure 9), refuting the designation of these cells as interstitial macrophages “IM”, which was a designation made largely in a single paper (Huang et al), and based largely on CD64 and MerTK staining. I believe the point could be made more simply and, again as a subpoint to the fact that these cells are not mostly DCs. The discrepancy in the % of the cells that are MerTK+ (the primary difference between this paper and that of Huang et al.) could perhaps be explained by the dose/route of infection (50-100 CFU by aerosol in this paper vs. 1000 CFU intranasally by Huang), but the biological significance of what % of the macrophages are MerTK+ is not clear and should be de-emphasized.

4) The manuscript uses several different designations throughout the manuscript in reference to these major infected cell type. Initially they are called *CD11bDCs and then switch at various times to calling them different names, including between “CD11c hi MC” and “MDC”. It is understandable that they initially call them CD11bDC as this is what they have been called historically, but the meaning of * is only in the figure legend and thus confusing. However, I believe they should settle then settle on the terminology they want to use and use it consistently thereafter.

5) In the introduction to Figure 2 on p.8, the authors state that “prior studies that identified cell types infected by Mtb using definitions based mostly on CD11b and CD11c, focused on one or two cell types, or looked at early timepoints” and cite several references for this statement. However, ref 9 (Cohen et al) is omitted in this section. Figure 1 in Cohen et al., used a panel of several markers, looked at several myeloid cell types, and arrived at the same nomenclature used here at similar timepoints. Thus, Figure 2 is consistent with and confirms this prior work and this should be acknowledged.

6) The authors use a dump gate of CD19, Thy1, and NK1.1 to exclude B cells, T cells, and NK cells from the myeloid populations they want to look at. These markers are usually lineage specific under non-inflamed conditions, but are the authors sure that these markers are not upregulated on myeloid cells during pulmonary TB? In Fig. S2 they argue that the dump is necessary because B cells/T cells/NK cells upregulate myeloid markers during Mtb infection. However, this same data could also be explained if myeloid cells upregulate CD19, Thy1, or NK1.1.

7) In Fig. 3 the authors show that AM were activated by IFNg and on p. 9 they state, based on this data, that “we infer that T cells are specifically interacting with Mtb-infected AM.” However, I believe it is possible that these AM were activated by bystander IFNg in the absence of a cognate interaction, thus this statement should be softened. To make this statement the authors would need to do MHCII-/- mixed chimeras and show a difference in IFNg activated genes in MHCII+ vs. MCHII- AMs.

8) It is unclear how the predicted pathways were determined in Fig 4D – was this using a bioinformatics approach or purely plotting known stimuli that could be involved in activating macrophages? The graph needs to be explained more clearly. Do the bars represent relative expression or p-value?

9) In Fig 4E-G, the Ingenuity platform used to assess the transcriptional data. I found this graph confusing. I am not very familiar with this platform, and perhaps that is the main problem, but perhaps other readers will also be unfamiliar and I believe it needs more explanation. For example, does Fig. 4F indicate that there are no pathways predicted to be significantly upregulated by infection in the MDC population? This seems insconsistent with Fig4B and 4H, which show several Th1-related genes upregulated? IL1b downregulations is highlighted in the text, but IL-1a and IL-12 are both upregulated in the 4B scatter plot. Please clarify.

10) In Figure 5, 25 genes were shown to be differentially expressed by AM, RM, and MDC that were infected, compared to those that were uninfected. The authors then went on to validate protein expression for three of these genes using antibodies (Abca1, CD38, and CD14) and investigate infected cells expressing these 3 markers for the rest of the paper. However, it is not clear how these 3 markers were chosen from amongst the 25, as the expression levels of these 3 genes were relatively low compared to others. Were other validations also attempted? Or were these 3 just the best antibodies? Without an explanation the designation of Triple and double positive cells in the remainder of the manuscript seems somewhat arbitrary.

11) The importance of the BMDC experiments in Fig8A-B is unclear, even as stated by the authors themselves, and may not be needed in the manuscript. What do the CD38-negative cells represent? Are these cells that just haven’t fully differentiated in the in vitro culture? Is the left-hand plot in Fig8A the uninfected controls?

12) Do the authors have any insights about why, in Fig. 8D ABCA1 expression goes down with IFNg in vitro (unlike CD38 and CD14) even though it’s highly expressed in vivo?

13) Figure 7D depicts NOS2 expression by various myeloid populations based Boolean gating of the markers ABCA1, CD38, and CD14. A comparison of NOS2 staining in Mtb infected AM vs. MDC vs. RM would also be interested. The authors make conclusions about exposure of all of these myeloid populations to IFNg, but this conclusion is based on RNA-Seq of sorted bulk populations of infected cells. These populations may include subpopulations that were and were not exposed to IFNg. Assessing NOS2 at a single cell level would be informative, as the % positive may vary and this may mean some populations contain cells that are less exposed to IFNg than others. If so, the conclusions about the bulk population being exposed to T cells and IFNg should be softened, as there may be subpopulations, even perhaps large subpopulations that are not.

14) The authors use a YFP-expressing Mtb strain that was transformed using a plasmid that expresses YFP driven by a constitutive promoter. Usually plasmids are lost over time and at late timepoints most infected cells no longer express the plasmid and are fluorescent. Thus, the utility of these strains are usually restricted to the first few weeks of infection. In Figure 9 and 10 this strain is used to identify infected cells even 16 weeks post-infection. What percentage of the colonies continue to be fluorescent at each of these timepoints (3 weeks, 6 weeks, and 16 weeks)? (In Figure 10 there seems to be a much lower % of YFP+ cells at 10 weeks than at 6 weeks, whereas overall CFUs usually don’t drop in this timeframe, suggesting plasmid loss). Unless the authors can convincingly show that the plasmid is not preferentially lost in some cell types relative to other cell types (which may be true if the Mtb replication rate is different in different cell types), I suggest that the authors not show data at timepoints in which the % of infected cells that still express YFP is low, as it would be difficult to draw conclusions.

15) The conclusions on p16 regarding AM, and there highly activated state should be softened considerably. In this context, the work of Rothchild et al should be recognized which shows that AM are very slow to upregulate pro-inflammatory pathways during the first 10 days of infection. It should also be acknowledged that the conclusions from bulk RNA-Seq could mean that a subset of AM are activated by IFNg, whereas others subpopulations may not. This would need to be evaluated on a single cell level, preferentially in tissue sections to identify the relative locations of activated and unactivated cells.

Reviewer #3: No major revisions

**Part III – Minor Issues: Editorial and Data Presentation Modifications**

Reviewer #1: (No Response)

Reviewer #2: 1. A lot of the FACS plots are missing percentage values inside the gates (for example, Fig 2C), which would be really helpful to interpreting the data. Also, in Figure 1H, should the AM percent say “85” instead of “12”?

2. The gating strategy shown in Fig S1 does not mention the “MDC” cells and instead only refers to them as CD11b DC. While you address this later on, it would be helpful for future figures if the MDC was mentioned there. Conversely, in Fig4H you again refer to CD11b DC instead of MDC.

3. The plots in Fig 4E-G should be labeled with the respective cell type for clarity.

4. Please include units for the Y-axis of Fig 4H. What do the bar heights represent? Are AM showing significantly stronger expression profiles for inflammatory genes compared to MDC? If so, this would conflict with Huang et al who find that IM (MDC) are more proinflammatory relative to AM.

5. Is IL-1b staining shown in Figure 7 is actually pro-IL1b. If so, please label it the plots and Methods accordingly.

6. Similar reporting of enriched NOS2 expression among Mtb-infected cells was reported in Huang et al and of enriched IL-1b in Cohen et al. Please acknowledge these when discussing Fig 7A. This activation status appears to be a general feature of Mtb-infected cells.

Reviewer #3: Minor revisions and thoughts:

Are Alveolar macrophages able to degrade the YFP protein more effectively than CD11chi MDC thus leaving unstained bacteria residing in these cells? Could an acid fast stain be done on these cells to determine they have lower numbers of bacteria.

Is there any evidence that CD11cHi MDC or equivalent cells exist in humans?

The authors describe the TP CD11cHi MDC having the highest level of NOS and IL-1beta which are normally associated with an increased bacillary killing. However Mtb seems to preferentially infect and reside in these cells how is this congruent ?

Alterations/Additions to figures.

Figure 1I. The first panel of this subsection examining Non PMN myeloid cells is a dot plot while the rest are contour plots.

Figure 4 Typo in legend of H

Figure 6 This figure lacks clarity specifically the A to H sub groups.

Figure 7 Same as figure 6

Figure 8D Statistics need to be applied to these data sets.

Figure 8E Collated MFI for ABCA1 between WT and KO should be graphed for figure 8E

Figure 10 Gates and numbers needed for this figure. Also the data should be collated and presented in this figure.

PLOS authors have the option to publish the peer review history of their article (what does this mean?). If published, this will include your full peer review and any attached files.

Reviewer #1: No

Reviewer #2: No

Reviewer #3: No
---

## [Editor Report · Decision Letter 1]

12 May 2020

Dear Dr. Behar,

We are pleased to inform you that your manuscript 'CD11cHi monocyte-derived macrophages are a major cellular compartment infected by Mycobacterium tuberculosis' has been provisionally accepted for publication in PLOS Pathogens.

Best regards,

Marcel A. Behr

Associate Editor

PLOS Pathogens

Michael Wessels

Section Editor

PLOS Pathogens

Kasturi Haldar

Editor-in-Chief

PLOS Pathogens

orcid.org/0000-0001-5065-158X

Michael Malim

Editor-in-Chief

PLOS Pathogens

orcid.org/0000-0002-7699-2064
---

## [Editor Report · Acceptance letter]

10 Jun 2020

Dear Dr. Behar,

We are delighted to inform you that your manuscript, "CD11c^Hi^ monocyte-derived macrophages are a major cellular compartment infected by *Mycobacterium tuberculosis*," has been formally accepted for publication in PLOS Pathogens.

Best regards,

Kasturi Haldar

Editor-in-Chief

PLOS Pathogens

orcid.org/0000-0001-5065-158X

Michael Malim

Editor-in-Chief

PLOS Pathogens

orcid.org/0000-0002-7699-2064